# Localization and Functional Roles of Components of the Translation Apparatus in the Eukaryotic Cell Nucleus

**DOI:** 10.3390/cells10113239

**Published:** 2021-11-19

**Authors:** Zaur M. Kachaev, Sergey D. Ivashchenko, Eugene N. Kozlov, Lyubov A. Lebedeva, Yulii V. Shidlovskii

**Affiliations:** 1Department of Gene Expression Regulation in Development, Institute of Gene Biology, Russian Academy of Sciences, 119334 Moscow, Russia; k-z-m@mail.ru (Z.M.K.); ivashchenko.sd@phystech.edu (S.D.I.); ugin.sfu@gmail.com (E.N.K.); ll78@yandex.ru (L.A.L.); 2Center for Genetics and Life Science, Sirius University of Science and Technology, 354340 Sochi, Russia; 3Department of Biology and General Genetics, Sechenov First Moscow State Medical University (Sechenov University), 119992 Moscow, Russia

**Keywords:** translation factor, ribosomal protein, gene expression, nucleus, stress, cell response, transcription, mRNA export, cancer, moonlighting protein

## Abstract

Components of the translation apparatus, including ribosomal proteins, have been found in cell nuclei in various organisms. Components of the translation apparatus are involved in various nuclear processes, particularly those associated with genome integrity control and the nuclear stages of gene expression, such as transcription, mRNA processing, and mRNA export. Components of the translation apparatus control intranuclear trafficking; the nuclear import and export of RNA and proteins; and regulate the activity, stability, and functional recruitment of nuclear proteins. The nuclear translocation of these components is often involved in the cell response to stimulation and stress, in addition to playing critical roles in oncogenesis and viral infection. Many components of the translation apparatus are moonlighting proteins, involved in integral cell stress response and coupling of gene expression subprocesses. Thus, this phenomenon represents a significant interest for both basic and applied molecular biology. Here, we provide an overview of the current data regarding the molecular functions of translation factors and ribosomal proteins in the cell nucleus.

## 1. Introduction

mRNA translation is performed by a large number of highly conserved proteins and RNAs. The main components of the translation apparatus (CTAs) include translation factors, ribosomal proteins (RPs), and ribosomal RNAs (rRNAs) [1]. The eukaryotic 80S ribosome consists of two subunits, the 40S and the 60S. The large subunit contains three types of rRNA (28S, 5.8S, and 5S) and approximately 50 RPL proteins, whereas the small subunit consists of a single 18S rRNA and approximately 30 RPS proteins. The ribosome components are encoded by hundreds of copies of rRNA genes and dozens of RP genes, and most of the ribosome biogenesis steps occur in the nucleolus [2,3,4]. Ribosome biogenesis is an energy-consumptive process that is finely regulated by various signaling pathways, including the target of rapamycin (TOR) pathway [5,6,7,8,9].

The core group of proteins that govern translation includes initiation, elongation, and termination/release factors. Translation initiation factors are highly conserved and include the eukaryotic initiation factors (eIFs) eIF1, eIF1A, eIF1B, eIF2 (a complex consisting of three subunits; eIF2S1 is also known as eIF2α), eIF2A, eIF2B (a complex consisting of five subunits), eIF3 (a complex consisting of up to 13 subunits), eIF4A, eIF4B, eIF4E, eIF4G, eIF4H, eIF5, eIF5B, and auxiliary eIF6. The eukaryotic elongation factors (eEFs) include eEF1A, eEF1B (a complex consisting of two to four subunits), eEF2, eIF5A, and eEFSec. The eukaryotic release factors include eRF1 and eRF3. Some CTAs have several paralogs and isoforms. The functioning of the core apparatus is accompanied by other proteins, in addition to these factors [1,10,11,12,13,14,15,16].

## 2. Nuclear and Subnuclear Localization of Translation Factors and Its Regulation

Initially, translation factors were thought to localize in the cytoplasm. In the 1990s, one of the first studies by N. Sonenberg reported that eIF4E was localized to the nuclei of COS-1 monkey cells [17], which was later found to be a conserved feature of this protein in other species [18,19]. Since then, hundreds of papers have described the nuclear localization of various CTAs, although in many cases, the functional roles associated with nuclear localization remain unclear and poorly explored.

Multiple screening studies have applied cell imaging or biochemical purification techniques to reveal the presence of translation factors in the nucleus. A localization screen of the YFP-tagged *Saccharomyces pombe* ORFeome revealed the nuclear or dual nuclear and cytoplasmic localization of eIF1A, eIF2S1, eIF2B1, eIF2B4, eIF3d, eIF3g, eIF3i, eIF5A1, eIF6, eEF1A-B, and eRF1 [20]. By contrast, an analysis of the GFP-tagged proteome in *Saccharomyces cerevisiae* revealed that only eIF6 was localized to the nucleus [21], although the nuclear localization of eIF4E in *S. cerevisiae* was described in a separate study [18]. In a screen of YFP-tagged *Drosophila* embryonic proteins, eIF1A showed nuclear localization [22].

The analysis of the nuclear proteome of *Arabidopsis* revealed the presence of eIF2S1 and several RPs in the nucleus. eIF2S1 and a set of RPs showed nuclear accumulation following cold stress, whereas another set of RPs showed the opposite behavior [23]. Another screen examining *Arabidopsis* protein localization identified the nuclear localization of eIF3f, eIF4E, eIF5, and eEF1A [24]. A recent analysis of the *Arabidopsis* nuclear proteome revealed additional CTAs with nuclear localization, and several RPs were found to be imported into the nucleus during the immune response [25,26].

Systematic immunostaining analysis of human translation factor localization revealed the presence of eIF1, eIF1A, and eIF6 in the nucleus [27]. eIF1A was also found in the nucleus of human cells following the inhibition of importin-13 [28]. The human protein haponin (eIF1AD), which is paralogous to eIF1A, resides in the nucleus and is involved in the oxidative stress response [29]. eIF6 was found in the human nucleolus, where it participates in ribosome biogenesis [30]. The elongation factor eEFSec presents a dual nucleocytoplasmic localization and utilizes CRM1 (exportin 1)-mediated nuclear export pathway [31]. In *Xenopus* oocytes, the nucleoproteome includes eIF4E, eIF6, and eEFSec [32]. A recent analysis of the chromatin-associated proteome in mouse embryonic stem cells revealed the presence of an almost complete set of translation factors and RPs [33].

Several translation factors have been found in specific subnuclear compartments, including the nucleolus, nuclear bodies, and speckles, or colocalizing with specific proteins. eIF4A1, eIF6, and eEF1A were found in the nucleolus in *Arabidopsis* [34]. eRF1 was found in the nucleolus in yeast, where it likely participates in the quality control mechanisms of maturing ribosomes [35]. eIF3k directly interacts with the nuclear promyelocytic leukemia protein (PML), and its isoform eIF3k2 is localized in PML bodies in human cells [36]. In addition, eIF3k interacts with and partially colocalizes with cyclin D3 in the nuclei of human cells [37]. eIF3e was found in nuclear bodies in human primary lymphocytes [38]. eIF4E was found in nuclear speckles, which often colocalize with PML bodies in human cells [39].

Fractionation techniques have further confirmed the association of CTAs with subnuclear structures. The analysis of interchromatin granule cluster compositions from mouse liver nuclei revealed the presence of eIF4A and multiple RPs [40]. The sequential extraction of proteins from human cells to obtain a chromosome scaffold fraction revealed several RPs in this fraction [41]. Another study of human cells showed the presence of eIF4A1, eIF5A1, eIF6, eEF1A1, eEF1Bγ, eEF1ε1, eEF2, and multiple RPs in the insoluble nuclear fraction, which contains nuclear architectural components [42].

Some of the interactomes reported for nuclear proteins also imply a nuclear localization for several CTAs. For example, the interactome reported for murine nuclear phosphoinositide-specific phospholipase Cβ1b (PI-PLCβ1b), which is involved in the regulation of cell division, revealed several RPs and subunits of eIF2, eIF3, eIF4A, eIF4B, eIF5A1, eIF5A2, and eEF1A [43]. In *Drosophila,* the interactions of eIF1A with the transcription factor (TF) Ssb-c31a, eIF5A with TF Mes2, eEF1Bδ and eEF2 with the histone-associated factors nucleoplasmin and Df31 have been reported [44].

Certain CTAs could be components of several protein complexes and thus have specific nuclear functions. eIF3e/Int6 is a subunit shared with the proteasome and COP9 signalosome, which are involved in protein degradation [45,46]. In yeasts, eIF3e is required for a proper accumulation of proteasomes in the nucleus, which is crucial for the progression of the cell cycle [47,48]. eIF3e and eIF3c are also associated with the COP9 signalosome in *Arabidopsis* nuclei [49].

Several molecular mechanisms are employed in the cell to control nuclear localization of translation factors. The phosphorylation status appears to affect the localization. Mammalian eIF2S1 was found in the nucleus, and its localization depends on the phosphorylation [50,51,52,53]. An analysis of the nuclear phosphoproteome in HeLa cells revealed the presence of phosphorylated eIF2B5, eIF4B, eIF4G1, eIF4G2, eIF3b, eIF3c, eIF3g, eIF5B, RPS3, and RPLP0 [54].

The nuclear accumulation of eIF5A1 and eIF5A2 paralogs is regulated via acetylation, which is supervised by nuclear PCAF, HDAC6, and Sirtuin enzymes. The mutually opposite modification by hypusine determines their localization in the cytoplasm in mammalian cells [55,56,57]. eIF5A2 was also found in the nuclei of *Arabidopsis* cells [58,59]. In plants, phosphorylation by casein kinase 2 (CK2) plays a role in the regulation of eIF5A nucleocytoplasmic shuttling [60,61]. In mammals, the nuclear entry of eIF5A1 has been initially suggested to occur via passive diffusion [62], but an NLS was later identified in the N-terminus of the protein [63]. eIF5A1 export occurs with the help of exportin-1 or exportin-4, whereas eIF5A2 export requires exportin-4 [57,64].

Other proteins may control the intracellular localization of translation factors. The transport of human eIF4E to the nucleus depends on the eIF4E transporter 4E-T, a nucleocytoplasmic shuttle protein that contains an eIF4E-binding site. In the presence of a 4E-T transporter, the formation of an importin-αβ heterodimer in a complex with eIF4E is possible [65,66]. Interestingly, in the nuclei of human cells, 4E-T colocalizes in PML-associated foci with another multifunctional factor, Pat1, which is also involved in translation control [67]. 4E-T also mediates the nuclear import of the translational repressor DDX6 in human cells [68] and serves as a general translation repressor [66]. In *Drosophila,* eIF4E interacts with the translation regulator cup, which is a shuttling protein, and this interaction is important for cup retention in the cytoplasm of ovarian cells [69].

Viral infection is one of the factors that affect the intracellular distribution of various CTAs. A fraction of eIF3e was found in PML bodies under normal conditions, whereas the binding of the human T-cell leukemia virus (HTLV-I) regulatory Tax protein with eIF3e causes its redistribution to the cytoplasm [70]. Contrary, eIF4A1 translocates to the nucleus and cooperates with the viral protein Rev to promote further Gag protein synthesis during HIV-1 replication in human cells [71]. Viral infection causes the strong nuclear accumulation of eIF4G in HeLa cells [72].

In addition to the core CTAs, other translational factors and translational regulators have been identified in the nucleus. The translation factor SLIP (MIF4GD), which is required for the replication-dependent translation of histone mRNAs, was found in both the nucleus and cytoplasm in human cells [73]. The translational repressor nanos3 was found in the nuclei of murine and human primordial germ cells [74,75]. The mTOR kinase, which acts as a general regulator of translation, was found in cell nuclei and has been associated with nuclear regulatory functions in human and murine cells [76,77]. The eIF2α (eIF2S1) kinase 2 PKR was also found in the nuclei of acute leukemia cells [78].

## 3. Regulation of RP Nuclear Localization

RPs enter the nucleus to participate in rRNA maturation and ribosome assembly [79,80,81], and RPs are abundant in the nucleolus. Indeed, study of the interactome of the nucleolar protein Nop132 [82] and direct nucleolar proteome isolation revealed multiple RPs [83]. Moreover, RPL11 and RPL15 are significant contributors to the integrity of the nucleolar structure in human cells [84]. RPs feature a nuclear localization signal (NLS), which is commonly found in highly conserved rRNA-binding domains and appears to be involved in rRNA folding [85]. Other eukaryotic-specific sequences in RPs have also been identified as involved in the nuclear trafficking of RPs [86]. NLSs of several RPs define their localization not only in the nucleuolus, but also in the nucleoplasm [87,88].

The various regulatory pathways and protein modifications mediate the nuclear and subnuclear localization of RPs [80,89,90,91,92]. The mTOR signaling pathway regulates the nuclear import of RPs in human cells [93]. RPL10B relocates to the nucleus upon UV irradiation in *Arabidopsis* [94]. The proper localization of RPS10 in the granular component of the nucleolus in human cells requires arginine methylation by protein arginine methyltransferase 5 (PRMT5) [95], whereas RPS3 transport to the nucleolus is dependent on arginine methylation by PRMT1 [96]. RPL3 in human cells is a substrate of nuclear methyltransferase-like 18 (METTL18); this modification is important for its role in ribosome biogenesis [97]. Modification by the small ubiquitin-like modifier protein (SUMO) regulates the nuclear localization of RPL22 in *Drosophila* meiotic spermatocytes [98].

Interaction with other molecules might affect the RP localization. Epstein–Barr virus (EBV) infection causes the relocalization of RPL22 in B lymphocytes via interactions between RPL22 and non-coding RNA [99,100]. The potato virus A causes the accumulation of several RPs in the nucleus [101]. By contrast, the rabies virus phosphoprotein interacts with RPL9, causing translocation from the nucleus to the cytoplasm in human cells [102]. The nucleolar localization of RPS9 in human cells depends on its interaction with the multifunctional protein nucleophosmin [103,104]. The tumor suppressor BCCIPβ interacts with RPS7 and modulates its extraribosomal functions in the nuclei of human cells [105]. BCCIPβ also forms a complex with nuclear RPL23 and eIF6 in human cells; this interaction stabilizes the nuclear pool of RPL23 [106]. The properties of a nuclear RPs could differ from those of their cytoplasmic counterparts; for example, not cytosolic but nuclear RPS3A specifically binds phosphatidylinositol trisphosphate second messenger in human cells [107].

RPs also contribute to the intranuclear localization of other proteins. RPL5 is abundant in the nucleolus in human cells, and its interaction with the nucleolar ATPase NVL2, which is involved in ribosome biogenesis, determines the specific nucleolar localization of NVL2 [88]. RPS3 or RPL19 are required for the nuclear localization of ERH protein in *Drosophila* [108]. RPL9 participates in the intranuclear trafficking of mammary tumor virus (MMTV) Gag protein [109]. RPL11 interacts with the nuclear protein PML and is important for the nucleolar localization of PML, which is induced by stress conditions in mammalian cells [110]. RPS6 interacts with the latency-associated nuclear antigen (LANA) of Kaposi’s sarcoma-associated herpesvirus, contributing to its stability [111].

A component of the 40S ribosomal subunit RACK1 is constitutively present in the nucleus of murine cells, where it is recruited into the BMAL1 complex in a circadian manner during the negative feedback phase of the circadian cycle [112]. RACK1 localization in the nucleus and interactions with eIF6 in both the cytoplasmic and nuclear compartments were also described in *Arabidopsis* [113].

CTAs have been found to be involved in multiple nuclear functions. We describe their key nuclear activities below.

## 4. Roles for CTAs in DNA Repair, Synthesis, and Genome Integrity Control

Several CTAs are essential components of the DNA damage response pathway. RPS3 is one of them. In response to genotoxic stress, RPS3 is translocated to the nucleus and recruited to the DNA damage loci in human cells. The translocation of RPS3 in the DNA damage response pathway is mediated by RPS3 phosphorylation due to extracellular signal-regulated kinase (ERK1/2), cyclin-dependent kinase (CDK1), and protein kinase C-δ [114,115,116,117], whereas nuclear protein phosphatase 2A appears to counteract this process [118]. RPS3 possesses endonuclease activity [119,120,121,122] and interacts with the base excision enzymes 8-oxoguanine DNA glycosylase 1 (OGG1) and apurinic/apyrimidinic endonuclease (APE/Ref-1) [123], in addition to uracil-DNA glycosylase (UNG) in human cells [124]. Human RPS3 additionally interacts with transcription factor IIH (TFIIH) and is involved in the nucleotide excision repair pathway [125]. RPS3 has also been identified to participate in DNA repair and replication through stress-induced interaction with RecQ-like helicase 4 (RecQl4) in human cells [126].

In addition to participating in DNA repair and sustaining the genome integrity, RPS3 can induce apoptosis. RPS3 binds E2F1 and promotes proapoptotic gene induction in rat neurons. However, Akt-dependent phosphorylation of RPS3 disrupts this interaction, stimulating the nuclear accumulation of RPS3, and its repair function in the nucleus sustains neuronal survival [127]. The translocation of RPS3 to the nuclear membrane in murine lymphocytic cells has also been associated with the induction of apoptosis [128].

Other RPs are also involved in DNA damage pathways. In the nucleolus, human RPSA interacts with RNF8 protein, which is involved in the DNA damage response. DNA damage causes the release of RNF8 and BRCA1 to the nucleoplasm, which is regulated by RPSA [129]. Human RPL6 interacts with the histone H2A/H2AX and is recruited to DNA damage loci in a poly (ADP-ribose) polymerase (PARP)-dependent manner. RPL6 is important for the binding of mediator of DNA damage checkpoint protein 1 (MDC1) with γH2AX and the further recruitment of additional repair proteins. RPL8 and RPS14 are also recruited to DNA damage sites [130].

Human RPS27L binds to proteins FANCD2 and FANCI, which are components of the interstrand cross-link repair pathway. RPS27L binding prevents their degradation and stimulates DNA repair [131]. Interaction in human cells has been reported between RPLP0 and the DNA repair enzyme and transcriptional co-activator APE1/Ref-1, which serves as a master regulator of the cellular response to oxidative stress conditions [132]. Moreover, RPLP0 has been hypothesized to act as an endonuclease involved in DNA repair in *Drosophila* [133].

The DNA repair and telomere maintenance protein nibrin (NBS1) might be regulated by Mdm2; their interaction is affected by several Mdm2-binding RPs in human cells [134] (see below). A role of human RPL3 in the control of DNA repair activity has also been described [135]. Human eIF2 participates in the stabilization of the DNA-dependent protein kinase (DNA-PKcs)–Ku complex during DNA double-stranded break repair, and eIF2S2 is a substrate of DNA-PK [136].

eIF3e localizes to DNA damage loci and participates in repair processes via interactions with ATM protein kinase to promote the loading of the RAD51 recombinase in human cells [137,138]. The COP9 signalosome, which carries eIF3e as a subunit, plays a regulatory role in the DNA repair response [139].

Several CTAs contribute to the regulation of DNA replication. RPL5A and RPL5B participate in regulating the telomere length set point in *Arabidopsis* [140]. Human eIF3e also interacts with the polyubiquitinylated form of the replication factor MCM7 in the nucleus, which prevents its proteasomal degradation and increases its association with chromatin [141]. RPL4 is essential for replication of viral DNA, acting via interactions with Epstein–Barr virus nuclear antigen 1 (EBNA1), and the formation of the origin of plasmid replication (oriP) complex [142].

The eEF1ε1 subunit (AIMP3, aminoacyl tRNA synthetase complex component) is translocated to the nuclei of actively proliferating human cells during the S-phase. AIMP3 also localizes to the nucleus in response to DNA damage by UV exposure and adriamycin treatments. In response to DNA damage, this factor directly interacts with the ATM/ATR kinase, resulting in the subsequent activation of p53 [143]. The deletion of AIMP3 in mice causes the accumulation of DNA damage, indicating its involvement in the regulation of genome stability [144,145]. Another component of the tRNA synthetase complex, AIMP2, contributes to the DNA damage response by translocating to the nucleus, interacting with p53, and preventing its Mdm2-mediated degradation in murine cells [146]. Nuclear AIMP2 also promotes the degradation of the FBP, a transcriptional activator of c-Myc in human cells [147].

Nuclear eEF2 has the opposite genome-destabilizing effect. It is phosphorylated by C-terminal Src kinase (CSK), which is coupled with proteolytic cleavage. The nuclear translocation of the cleaved product causes genome instability, nuclear deformation, and aneuploidy formation in human cells [148].

## 5. Roles of CTAs in Transcriptional Regulation

RPs and rRNAs have been identified in active transcription sites. The recruitment of RPs to chromatin occurs co-transcriptionally in an RNA-dependent manner. RPS2, RPS5A, RPS9, RPS11, RPS13, RPS18, RPL8, RPL11, RPL32, and RPL36 have been found at active loci on polytene chromosomes in *Drosophila*, and RNase treatment substantially reduces their association with chromosomes. In addition, eIF5B and eRF3 have been found at active transcription sites on polytene chromosomes, colocalizing with RNA polymerase II (RNAP II) [149,150].

The presence of RPL7, RPL11, and RPL25 (homolog of human RPL23A) at active transcription loci has been observed in *S. pombe*; RPs localize to protein-coding and non-protein-coding genes, including tRNA, snoRNA, snRNA, and 5S rRNA genes. Moreover, these RPs are also localized on genomic repeats and centromeres. RP recruitment to chromosomes in yeast occurs predominantly in an RNA-dependent manner [151]. The association of RPS7, RPL7, RPL26, and RPL34 with nascent transcripts in yeast was shown in another study [152]. RPS14 inhibits the transcription of its own gene in human cells [153]. RPL12 is required for the transcription of phosphate signal transduction (PHO) pathway genes in yeast [154].

The presence of CTAs in condensed chromatin has also been described. Several RPs interact with histone H1 in *Drosophila*, and H1 and RPL22 reside in condensed chromatin. RP–H1 interactions are likely important for transcriptional repression [155]. The analysis of the H1 interactome revealed multiple RPs, eIF3 subunits, and other CTAs [156,157]. The interactome for the heterochromatin protein HP1a in *Drosophila* includes eIF2S2, eIF3d1, eIF4A, eIF4B, eIF5A, eIF4G, eEF1A1, eEF2, and multiple RPs [158], and eIF4A interacts with HP1c in *Drosophila* [159]. The nuclei of human sperm containing condensed chromatin are also abundant in multiple RPs [160].

Multiple interactions of RPs with specific and general TFs have been described. RPs are commonly involved in transcriptional regulation through the modulation of transcription factors (TFs) function. Several RPs were co-purified with TFIIIE and recruited to tRNA and 5S rRNA genes in *S. cerevisiae* [161]. RPL11 represses c-Myc–dependent RNAP III transcription in mammalian cells [162], and RPS20 is involved in RNAP III transcription termination control in yeast [163]. In mammals, RPL11 binds to the c-Myc and represses the activity of its target promoters [164]. Human RPS14 also interacts with c-Myc and prevents the recruitment of c-Myc and its co-activator TRRAP to target genes [165]. Nuclear RPL3 binds to the phosphorylated TF Sp1, which is hypothesized to result in promoter-dependent effects, resulting in either the dissociation or stable recruitment of Sp1 in human cells [166].

Murine RPS3A binds the TF C/EBP homologous protein (CHOP, also known as GADD153) to inhibit its activity [167]. Human RPS3A also inhibits the activities of the transcriptional co-activator EBNA5 of the Epstein–Barr virus [168] and nuclear PARP [169]. Human RPS2 binds the putative TF zinc finger protein 277 (ZNF277) in human cells [170]. RPLP1 and RPLP2 show intrinsic potential to activate transcription in yeast [171].

Mammalian RPL13 stimulates the activity of *NF-κB* and *IFN-β* promoters and is targeted by specific viral proteases due to its contributions to the antiviral response [172]. RPL10A in *Arabidopsis* relocates to the nucleus after phosphorylation by NIK1 kinase [173]. In the nucleus, RPL10A interacts with the transcription repressor L10-interacting MYB domain-containing protein (LIMYB), which downregulates the expression of RP genes as a component of the antiviral defense strategy in plants [174].

RPs can affect the recruitment of TFs to their target loci. RPL6 mediates the DNA binding of the TF Tax, expressed by HTLV-1 [175]. Human RPL7 counteracts the binding of vitamin D receptor retinoid X receptor (VDR-RXR) with its target loci [176], whereas rat RPL11 counteracts the binding of peroxisome proliferator-activated receptor-α (PPAR-α) [177]. RPL10 participates in the suppression of c-Jun homodimer binding with DNA in human cells [178].

In mammals, RPS3 is essential for nuclear factor (NF)-κB signaling by stabilizing the NF-κB binding with target genes [179]. Modification of the NF-κB p65 subunit promotes its binding with RPS3 [180], an interaction that is enhanced by the factor of immune response lipocalin 2 [181]. The nuclear localization of RPS3 requires phosphorylation by the inhibitor of NF-κB kinase (IKKβ) or casein kinase 2α, and nuclear RPS3 promotes specific NF-κB functions [182,183]. By contrast, the deubiquitination of human RPS3 blocks its nuclear translocation [184]. Human RPS3 also binds p53 to protect it from ubiquitination [185].

RPs are involved in the regulation of p53 transcriptional response. In mammals, multiple RPs bind to Mdm2, an E3 ligase and negative regulator of p53 [186,187,188,189,190,191,192,193]. The RP–Mdm2–p53 pathway connects ribosomal biogenesis with p53 activity [194]. Nucleolar stress causes the release of RPs to the nucleoplasm, which blocks Mdm2 and stimulates p53 activity. RPL11 and RPL5 are the principal players in this process [166,195]. Moreover, the formation of a complex between human RPL11 and Mdm2 is required for the recruitment of the p53 transcriptional coactivators p300/CBP to target promoters and the acetylation of p53 at K382. This process is accompanied by the neddylation of RPL11 [196]. This modification controls both nuclear and nucleolar localization of human RPL11, also contributing to the regulation of p53 activity [197,198]. Human RPL11 also directly interacts with the tumor suppressor ADP-ribosylation factor (ARF), forming a complex with Mdm2 and p53, which enhances p53 transcriptional activity [199]. The nucleolar protein GRWD1 mediates the opposite effect by binding RPL11 and blocking its interaction with Mdm2 in human cells [200]. Another nucleolar protein, spindling 1 (SPIN1), sequesters human RPL5 in the nucleolus, preventing its interaction with Mdm2 [201].

RPS26 interacts with p53 independently of Mdm2, forming a complex with p53 and p300, contributing to the p53 transcriptional response in mammals [202]. Genotoxic agents cause the proteasomal degradation of human RPL37 in the nucleoplasm and trigger the RPL11-dependent stabilization of p53 [203]. Similarly, silencing of human RPS9 activates p53 [204]. In addition, RPS2, RPS7, and RPS27A are substrates of Mdm2 in human cells, further contributing to the regulation of the p53 response [189,191,192,205].

RPs contribute to E2F1 functioning, as mentioned above for RPS3 [127]. RPL11 binding to Mdm2 stimulates E2F1 degradation and induces apoptosis in human cells [166,195]. RPL3 also negatively regulates the activity of E2F1-dependent promoters by interacting with PARP1 and sequestering it from promoters in human cancer cells [206].

Human RPL5 and RPL11 induce the transcriptional activity of TAp73, a p53 homolog. Upon ribosomal stress, the RPs bind the transactivation domain of TAp73 to block the binding of the Mdm2 negative regulator [207]. Human RPS7, an Mdm2 partner protein, interacts with the multifunctional GADD45α nuclear protein, preventing its Mdm2-mediated degradation. The cytotoxic agent arsenite enhances the RPS7–Mdm2 interaction, preventing GADD45α ubiquitination and degradation, inducing the activation of a GADD45α-dependent cell death pathway [208].

RPs also affect gene expression via epigenetic regulation. In *Drosophila*, the K3me3-modified RPL12 interacts with the epigenetic cofactor Corto via the Corto chromodomain. Corto and RPL12 colocalize in active chromatin on polytene chromosomes and are jointly involved in the regulation of euchromatin gene transcription, particularly heat shock genes and RP genes [209]. In this case, RPL12 might function jointly with the transcription factors Mad/Med, which are effectors in the BMP signaling pathway [210]. Another RP, RPLP0, is thought to be involved in epigenetic regulation by modifying position-effect variegation in *Drosophila* [211].

RPS6 is recruited to promoters of rRNA genes and, together with histone deacetylase 2, silences rDNA transcription in plants [212]. Plant RPS6 also interacts with nucleosome assembly protein 1 (NAP1), which, on the contrary, promotes rDNA transcription [213].

Direct interactions of several CTAs with RNA polymerases were found. Interactions between the Rpb11cα minor isoform of the human RNAP II subunit with eIF3a, eIF3i, and eIF3m have been described [214]. The Rpb4/7 heterodimer shows physical and functional interactions with the eIF3j and eIF3c subunits in yeast, although this interaction was investigated in the context of translation rather than transcription [215]. The RNAP II interactome in human cells shows the presence of eIF2S1, eIF4A1, eIF4A2, eIF4G, eIF5B, eIF6, eEF1A1, and multiple RPs. Although an extract from mitotic cells was used in this study, RNAP II interactions with other cytoplasmic proteins were much weaker than those with CTAs [216]. In line with these data, eIF3m depletion has a strong effect on transcription compared with a limited effect on translation in mice [217].

Mammalian eIF3l (PAF67) is found in the nucleolus and acts as a cofactor for RNAP I, participating in the initiation of rRNA gene transcription [218,219]. Phosphorylated eIF2S1 (eIF2α) is hypothesized to have direct effects on the transcriptional activities of the basal TF RRN3/TIF-IA and RNAP I transcription in human cells [220].

The direct participation of translation factors in transcriptional regulation has been described in multiple studies. The C-terminal domain of eIF3e increases the activity of Pap1 TF, a component of the oxidative stress response in yeast [221]. eIF3e also interacts with the transcriptional repressor Rfp, which colocalizes in PML-containing nuclear bodies in mammalian lymphocytes. Rfp drives the translocation of eIF3e to nuclear bodies [38].

eIF3f in the human cell nucleus regulates the expression of 34 genes (eIF3F gene cluster) via cooperation with signal transducer and activator of transcription 3 (STAT3) and other TFs [222]. In the nucleus, eIF3f interacts with and colocalizes with CDK11 in human cells, and phosphorylation by CDK11 is likely responsible for the nuclear localization of eIF3f [223,224]. eIF3h has also been suggested to play a role in transcription or epigenetic regulation and has been described as an enhancer of variegation in mice [225].

Nuclear eIF5A2 binds to the promoter of the *HIF1α* gene, activating its transcription. An accumulation and nuclear translocation of eIF5A2 in human cells are induced by hypoxia. [226].

eEF1A displays nuclear localization in human fibroblasts [227]. eEF1A is involved in the heat shock response through the eEF1A1-mediated stimulation of heat shock factor 1 (HSF-1), which is recruited to the *HSP70* gene promoter in human cells, resulting in *HSP70* transcription. eEF1A1 also binds to the elongating RNAP II and the 3′UTR of *HSP70* mRNA, contributing to the stabilization and export of mRNA from the nuclei. By contrast, the paralogous factor eEF1A2 does not affect HSF-1 binding to the promoter [228]. The interaction between RNAP II and TAR RNA of HIV-1 is stabilized by eEF1A, which is important for transcriptional stimulation [229]. Nuclear eEF1A in *Trypanosoma* has been suggested to be involved in specific transcriptional programs [230].

eEF1A also interacts with the zinc finger-associated domain (ZAD) of the TFs Zw5, ZIPIC, and Grau in *Drosophila,* presumably regulating the transcriptional activity of their target genes [231]. eEF1A in human T lymphocytes forms a complex with the tyrosine kinase Txk and PARP1. eEF1A and PARP1 are phosphorylated and translocated into the nucleus upon cell stimulation. This complex is recruited to the *IFN-γ* gene promoter and supports the transcriptional activity of *IFN-γ* [232]. eEF1A also interacts with zinc finger protein ZPR1 in both mammalian cells and yeast. ZPR1 is a signaling factor that communicates proliferative growth signals from the cytoplasm to the nucleus. Upon stimulation, both proteins are translocated into the nucleus, which is an important process for cellular proliferation [233,234]. In the murine cell nucleus, eEF1A2 is a substrate of PKCβI kinase, which is involved in various signaling pathways [235].

eEF1B has also been found in the nuclei of human cells. The interactome of nuclear eEF1Bβ indicates a putative role in transcription, splicing, and DNA damage response, whereas the interactome of nuclear eEF1Bγ suggests a role in the splicing and control of mRNA stability [236]. Moreover, eEF1Bγ binds the Rpb3 subunit of RNAP II and is recruited to the promoters of genes encoding vimentin, Che1 (AATF), and p53 in human cells [237,238]. The nuclear localization of eEF1Bγ has also been described in *Drosophila* [239]. A specific long isoform, eEF1BδL, is highly expressed in the human brain and testes. This isoform harbors an additional N-terminal sequence with an NLS, resulting in nuclear localization. This protein is a TF that cooperates with HSF-1 and Nrf2 TFs to support the transcription of heat shock element–carrying genes [240,241].

In addition to direct participation in transcription, CTAs regulate the subcellular/subnuclear localization and cell levels of specific TFs, thus indirectly affecting transcriptional activity. RPL23 serves as a negative regulator of Myc-associated zinc finger protein (MIZ-1)-dependent transcription in human cells by retaining its essential coactivator nucleophosmin in the nucleolus [242]. Human RPS27 is required for NF-κB phosphorylation and nuclear translocation [243]. RPS27 also regulates NF-κB signaling in shrimp [244]. Human RPS3A stimulates NF-κB nuclear translocation synergistically with hepatitis B virus X protein (HBx) [245]. RPL41 induces the phosphorylation and relocalization of the activating transcription factor 4 (ATF4) from the nucleus to the cytoplasm, resulting in its subsequent proteasomal degradation in human cancer cells [246]. Stress conditions induce eIF2S1 (eIF2α) phosphorylation, resulting in the general inhibition of translation. However, simultaneous activation of specific translation of the *ATF4* mRNA was described in mammalian cells. Increased levels of ATF4 induce a specific transcription program that allows the cell to respond to stress [247]. eEF1A participates in the phosphorylation and nuclear localization of the STAT3 TF upon *Helicobacter* infection in mammals [248]. eIF3e interacts with and directs the proteasomal degradation of HIF-2α in mammals [45,249]. Human eIF3f is a deubiquitinase that deubiquitinates the Notch1 receptor, allowing for its TF activity [250]. eIF3h deubiquitinases YAP and Snail TFs, which stabilizes these proteins and promotes the corresponding signaling in human cells [251,252].

eEF1A is a component of the nuclear protein export pathway in mammalian cells. Cargo proteins harboring specific transcription-dependent nuclear export motifs couple export with RNAP II transcription [253]. The signal for eEF1A-dependent export is a polyalanine tract, the disruption of which can result in the mislocalization of several TFs and disease development [254]. Acetylated eEF1A1 is translocated to the nucleus in mammalian nervous system cells, where it binds the TF Sox10 and promotes its export [255]. Human eEF1A is also involved in the nuclear export of the Snail TF through the Exp5-Aminoacyl-tRNA complex [256]. Mammalian eEF1A is exported from the nucleus via interaction with exportin-5, which is tRNA-dependent [27,257]. In yeast, eEF1A is also required for the re-export of aminoacylated tRNAs to the cytoplasm [258].

Human tyrosyl-tRNA synthetase (TyrRS) regulates gene expression by an epigenetic mechanism. Stress conditions cause the nuclear localization of TyrRS. The binding of nuclear TyrRS to TRIM28/histone deacetylase 1 (HDAC1) repressor complex blocks its activity toward E2F1 and stimulates the transcription of E2F1-dependent genes [259]. TyrRS also binds 20 genes encoding translation machinery components, recruits the TRIM28/HDAC1 or nucleosome remodeling deacetylase (NuRD) complex, and represses the transcription of these loci [260]. The nuclear translocation of TyrRS is regulated by acetylation, which is under control of p300/CBP-associated factor (PCAF) and sirtuin 1 enzymes [261]. Some mutations in TyrRS have been associated with E2F1 hyperactivation and the development of Charcot-Marie-Tooth neuropathy [262].

Cytoplasmic polyA-binding protein (PABPC) is a multifunctional RNA-binding protein that regulates various aspects of protein translation and mRNA stability. Several paralogous PABPCs have been described in mammals and plants; studies in mammals usually focus on PABPC1 as a predominant one in the cell. Nuclear translocation of PABPC is specifically induced by infection with viruses of various classes or occurs in response to cell stress in mammals and plants [263,264,265,266,267,268,269,270,271,272,273,274,275]. Virus-induced nuclear translocation of PABPC causes the general inhibition of translation [276] while allowing for viral protein synthesis to continue [277]. Viral protein synthesis could be realized via viral nuclear non-coding RNAs, which bind nuclear PABPC and selectively targets viral RNA for export and expression [278]. The nuclear accumulation of PABPC causes mRNA hyperadenylation and blocks its export, downregulating gene expression in mammalian cells [279]. In addition to the general translation shutoff, the nuclear localization of PABPC could result in the reprogramming of translation for specific subsets of mRNAs and affect mRNA stability and splicing [270,280].

The inhibition of transcription results in PABPC1 accumulation in nuclear speckles in human cells where it colocalizes with the splicing component SC35, and the nuclear localization of PABPC1 requires ATP [281]. In mammals, PABPC1 in the nucleus binds to polyadenylated intron-containing pre-mRNA [282]. An interaction between human PABPC1 and histone H1 was also described [157], and in virus-infected cells, fish PABPC is associated with both histones and splicing factors [276].

PABPC nuclear import depends on an interaction with α-importins, whereas the nuclear export of PABPC depends on ongoing mRNA export and occurs in an mRNA-dependent manner in mammals [280,283]. Export of mammalian PABPC1 occurs in a transcription-dependent manner with the participation of eEF1A [253]. Human PABPC1 participates in the nuclear trafficking of long interspersed element-1 (LINE1) ribonuclear protein (RNP), which is essential for L1 retrotransposition [284]. The NLS of PABPC resides in an RNA-binding domain and is masked by interactions with RNA. Thus, mRNA degradation or cytoplasmic deadenylation exposes the NLS, allowing for interactions with importins and driving the nuclear relocalization of PABPC [285,286].

The nuclear translocation of human PABPC impairs the recruitment of RNAP II and TATA-binding protein to promoters during the early stages of transcription but has no effects on the phosphorylation of the RNAP II C-terminal domain (CTD) or the recruitment of RNAP III. Thus, PABPC serves as an indicator of mRNA abundance, as the degradation of cytoplasmic mRNA results in its translocation to the nucleus and the inhibition of transcription [287]. As a result, PABPC has opposing effects on mRNA depending on its localization, stabilizes mRNA in the cytoplasm, and represses its synthesis in the nucleus. This mechanism represents an important feedback pathway that connects nuclear transcription with the cytoplasmic fate of mRNA [288,289].

Other proteins are also involved in the regulation of PABPC localization. The RNA helicase DEAD-box helicase 3 X-linked (DDX3X) interacts with PABPC1, and the downregulation of DDX3X causes the nuclear accumulation of PABPC1 in human cells [290,291]. A similar effect was observed for the cytoskeletal protein paxillin, which interacts with human PABPC1 in the nucleus, and both proteins are exported via the CRM1-exportin pathway [292].

PABPC1 binds the androgen receptor (AR) and is required for both AR-dependent transcription and its nuclear localization in human prostate cells [293]. Interestingly, paxillin is also a coregulator of AR, which is induced by androgens and EGF signaling [294]. PABPC stimulates the nuclear import of the TF Mld, but not other TFs during the ecdysone response in *Drosophila* [295].

For the single polyA-binding protein (PAB1) in *S. cerevisiae,* the nuclear import receptor Kap108/Sxm1 is known to be involved in the shuttling of PAB1 between the nucleus and the cytoplasm. Interestingly, the disruption of PAB1 import into the nucleus also affects mRNA export from the nucleus [296]. PAB1 is associated with cleavage factor IA (CFIA) and participates in the formation of the 3′-end of mRNA [297].

Participation in transcription has been described for *Drosophila* PABPC-interacting protein 2 (Paip2)*,* which is a translational inhibitor [298]. Paip2 was also found in the nucleus, where it binds active promoters in an RNA-dependent manner [299]. Paip2 is associated with various proteins, including the mRNA capping protein CBP80. Paip2 and CBP80 regulate the level of serine-5 CTD RNAP II phosphorylation on the promoter [300] and participate in the mRNA capping checkpoint [301].

## 6. Role of CTAs in mRNA Processing

Nuclear CTAs have been associated with mRNA splicing and 5′- and 3′-processing. RPs affect the transcription, splicing, and translation stages of their own gene expression [302,303,304]. In mammals, RPS13 represses its own expression by binding its pre-mRNA and disrupting spliceosome assembly to inhibit splicing [305]. The self-suppression of pre-mRNA splicing was also shown for RPS9 [306], RPS14 [307,308], RPS26 [309], RPL3 [310], RPL4 [311], RPL10A [312], RPL12 [313], RPL22 [314], RPL30 [315,316], and RPL32 [317,318] in various organisms. Regulation of RP gene expression via splicing is of special importance in budding yeasts, in which RP genes belong to a small group of intron-containing genes [306,319,320].

Human RPL3 is associated with alternative splicing by recruiting several factors to its own pre-mRNA to promote the selection of an alternative splice site, forming an mRNA with a premature stop codon that is degraded by the NMD pathway [310,321,322]. The pre-mRNA splicing of other genes is also regulated by these proteins. RPL22 and its paralog RPL22l1 have opposing effects on splicing in *Danio rerio* during gastrulation. RPL22 binds introns and induces exon 9 skipping in *smad2* pre-mRNA, whereas RPL22l1 promotes exon 9 inclusion [323]. Two RPL22 paralogs in yeast negatively affect the splicing of each other’s pre-mRNAs [324]. RPL26 participates in *p53* pre-mRNA splicing and the generation of specific mRNA isoforms in human cancer cells [325].

Human RPS3A interacts with the U5 and U11 snRNAs, which are required for U11 snRNA processing and minor-class splicing [326]. In *Arabidopsis*, RPL24 binds pre-miRNA and facilitates its processing [327].

Human eIF4E colocalizes with splicing factors (U1 snRNP) in cell nuclei and concentrates in nuclear speckles in an RNA-independent manner. Speckles appear to serve as functional sites for the formation of certain RNPs. Treatment of cells with the DRB (an inhibitor of RNAP II activity) causes the condensation of eIF4E granules, whereas overexpression of the Clk/Sty kinase causes dispersion of nuclear speckles [328]. eIF4E contributes to the female-specific alternative splicing of *msl*-2 and *Sxl* pre-mRNAs in *Drosophila*. eIF4E has been associated with several splicing factors in flies [329].

Human eIF4E interacts with factors involved in mRNA 3′ processing in the nucleus, stimulating the maturation of several transcripts in human cells, which occurs independently of its participation in mRNA export [330]. Human eIF4E also stimulates the capping of hundreds of coding and non-coding RNAs harboring a specific signal [331].

In human cells eIF2A, eIF4A, eIF4G1, eIF3a, eIF3b1, eIF3e, eIF3f, eIF3h, eIF3i1, eEF1A, and eEF1Bγ are associated with the pre-mRNA 3′ processing complex [332]. The yeast homolog of eEF1A, the Tef1 protein, interacts with the CFI pre-mRNA processing factor [333,334].

Human eIF3e co-purifies with nuclear mRNA cap-binding protein CBP80 [335].

Human eIF4A1 is also associated with the nuclear CBP80, and this interaction is further stimulated by the viral protein Rev during HIV-1 infection [71]. In *Drosophila*, nuclear eIF4A was also described [336], and its association with splicing regulators was shown [337]. The paralogous factor eIF4A3 is not involved in translation but serves as a component of the spliceosome and the exon junction complex (EJC) [338] and participates in ribosome biogenesis [339]. Interestingly, subcellular localization of eIF4A3 is also regulated [340,341].

A significant pool of the translation initiation factor eIF4G is localized in the nuclei of mammalian cells, where it interacts with the cap-binding complex (CBC) but not with nuclear eIF4E. eIF4G also interacts with spliceosomal snRNPs and splicing factors. eIF4G is recruited to pre-mRNA via the CBC, and this complex is exported to the cytoplasm [342]. In yeast, eIF4G is also associated with spliceosome components, and its depletion downregulates splicing [343].

eIF4G3 (but not the eIF4G1 paralog) localizes to the nuclei of mammalian spermatocytes in the region of a specific XY body. Mutations in the eIF4G3 gene cause meiosis arrest and male infertility. eIF4A1, eIF4E, eIF4E2, RPS6, RPL10L, and other CTAs are also localized in the XY body, where they are likely associated with mRNA metabolism [344].

Cytoplasmic polyadenylation element-binding (CPEB) proteins promote polyadenylation-induced translation in the cytoplasm [345,346], and all CPEB proteins act as shuttling proteins. The blockade of CRM-1–dependent nuclear export causes the nuclear accumulation of CPEB1, CPEB3, and CPEB4 in human cells. Stimulated neurons also show increased CPEB accumulation in the nucleus. In particular, the nuclear accumulation of CPEB4 depends on the depletion of calcium from the ER [347]. Nuclear CPEB1 in human cells was found in a few foci associated with the nucleoli, which contain CRM1 and might play a role in ribosome biogenesis [348]. In *Xenopus* oocytes, CPEB is associated with lampbrush chromosomes and several factors involved in nuclear RNA processing. In mouse fibroblasts, CPEB mediates alternative splicing [349]. In HeLa cells, CPEB1 colocalizes with splicing factors in the nucleus. CPEB1 binds with pre-mRNA, mediates the selection of alternative sites of polyadenylation, and shortens the 3′UTR of different transcripts. CPEB1 also affects alternative splicing by preventing the recruitment of the splicing factor U2AF65 [350]. During neurogenesis, murine CPEB3 serves as a crucial regulator of alternative splicing [351]. Nuclear CPEB3 interacts with STAT5b and inhibits transcriptional activity in mouse neurons. The NMDA-mediated activation of neurons stimulates the nuclear accumulation of CPEB3 in an IPO5-dependent manner, further downregulating the expression of the STAT5b target gene *EGFR* [352,353].

## 7. Role of CTAs in Nuclear mRNA Export

As indicated above, eIF4E is involved in nuclear mRNA biogenesis. Moreover, it defines a specific mRNA export pathway. In the nucleus, human eIF4E is specifically associated with the mRNA encoding the CDK1 and is involved in its export. The eIF4E-sensitive element (4E-SE) binding site for eIF4E in *CDK1* mRNA is approximately 100 nt found in the 3΄UTR [354]. In human cells, the overexpression of eIF4E alters the composition of the nuclear pore complex (NPC), increasing the export of *Gle1*, *DDX19*, and *RanBP1* mRNA [355].

The eIF4E-dependent export pathway for RNPs differs from the general RNP export pathways [TAP/NXF1 or REF/Aly] and requires the participation of the CRM1-exportin system. Currently, approximately 3000 transcripts in human cells are known to be eIF4E export targets, many of which encode oncoproteins [356,357,358].

eIF4E-dependent mRNA export is affected by other proteins. Human eIF4E interacts with PML, which reduces the affinity of eIF4E for the mRNA cap [359]. PML can cause the retention of *cyclin D1* mRNA in the nucleus, while eIF4E inhibits this retention and alters the morphology of PML bodies in human cells [39]. The human homeodomain protein HOXA9 stimulates the eIF4E-dependent export of mRNAs encoding cyclin D1 and ornithine decarboxylase (ODC) from nuclei, increasing the efficiency of ODC synthesis. This function of HOXA9 is transcriptionally independent and occurs via competition with another homeodomain protein PRH, which represses the eIF4E function [360]. Several candidate cofactors for the human eIF4E have been identified, which are associated with mRNA export [357,358]. One of these eIF4E partners is the LRPPRC protein, which binds both eIF4E and the 4E-SE element in mRNA. The overexpression of LRPPRC affects the nuclear export of several eIF4E-dependent mRNAs.

eIF5A is another factor that contributes to mRNA export control. eIF5A1 is associated with the intranuclear filaments of the NPC in mammalian cells and *Xenopus* oocytes. eIF5A1 acts as a shuttling protein, which interacts with the CRM1 nuclear receptor. In particular, eIF5A1 is essential for HIV-1 *Rev* RNA transport [361]. Moreover, nuclear eIF5A1 is important for HIV-1 Rev protein functions in transcriptional activation and viral RNA export [362,363,364]. eIF5A is additionally acts as a cofactor of HTLV-I Rex RNA export factor [365].

The role of the eIF5A-dependent export pathway was shown for several other transcripts. In mammalian cells, Sirtuin-1 (Sirt1) serves as a pH-sensor that deacetylates nuclear eIF5A during anaerobiosis, directing the export of eIF5A with associated *TSC2* mRNA. TSC2 induces metabolic depression [366]. Hypusinated eIF5A transports a set of specific mRNAs from the nucleus to ribosomes for translation, which is a mechanism employed by murine macrophages due to the induction of hypusinating deoxyhypusine synthase (DHPS) enzyme by bacterial infections [367]. Hypusination of eIF5A is crucial for the export of the *Nos2* mRNA upon the cytokine response of islet β cells in mice [368]. Hypusine modification of eIF5A is essential for *CD83* mRNA export and the full stimulatory activity of mature human dendritic cells [369].

The negative regulator of translation PDCD4, which controls eIF4A function during translation, resides predominantly in the nucleus under normal growth conditions in murine cells. Under serum starvation conditions, PDCD4 accumulates in the cytoplasm, although the inhibition of CRM1-dependent nuclear export prevents this accumulation. PDCD4 binds RNA and may be involved in nuclear RNA metabolism [370]. During metastasis in several cancers, a shift can be observed from a nuclear to cytoplasmic localization for PDCD4 [371].

## 8. Nuclear Localization of some CTAs Is Associated with Oncogenesis

The nuclear and subnuclear localization of several CTAs is associated with the development of an oncogenic phenotype. The RP–Mdm2–p53 pathway, described above, is important for tumor surveillance [194]. RPL11 retention in the nucleolus is important for tumor progression. The tumor suppressor PICT1 interacts with RPL11 and other RPs to maintain their nucleolar localization. Under stress conditions, PICT1 becomes depleted, and RPL11 escapes into the nucleoplasm, where it binds Mdm2 and blocks p53 ubiquitination [166,372,373]. The relocalization of RPL11 to the nucleoplasm can be detected under oncogenic or replicative stress conditions, and other components of the 5S RNP complex (RPL5 and 5S rRNA) are also involved in p53 activation [374].

The formation of an alternative Mdm2-RPL5 complex with the splicing factor SRSF1 also contributes to p53 stabilization and has been described in cells under stress conditions [375]. RPL26 regulates *p53* pre-mRNA splicing. Ionizing irradiation or methyl methanesulfonate treatment induces the binding of RPL26 to *p53* pre-mRNA, further inducing the recruitment of the splicing factor SRSF7 and the generation of alternatively spliced *p53β* mRNA, which can induce a cellular senescent phenotype [325]. In several types of cancer, RPS3 as a component of the NF-κB TF contributes to the upregulation of prosurvival genes, radioresistance, and cancer development [134].

Specific RNA giant nuclear body has been observed in cancer cells but not normal cells. eIF3d, eIF4A1, eIF4E, eEF1Bδ, eEF2, and 47 RPs contribute to the composition of the body [376]. Nuclear accumulation of eIF2S1 (eIF2α) in gastrointestinal carcinomas cells [377] and meningioma cells [378] has also been described.

The regulation of eIF3e nuclear localization could be a mechanism for tumorigenesis, as shown in fibroblasts [45,379]. eIF3f is localized in the nuclei of adenocarcinoma cells and regulates the expression of genes that control key events accompanying tumor formation. In particular, eIF3f regulates the expression of the central effector of metastases, *Snail2*, which is important for the induction of the epithelial-mesenchymal transition [222].

eIF4E is a pro-oncogenic protein that is highly upregulated in many cancers [380]. The abundance of nuclear phosphorylated eIF4E, such as in oligodendroglial tumors [378], is associated with increased tumor burden and reduced response to chemotherapy [381]. The overexpression of RAN binding protein 2 (RANBP2) specifically inhibits the eIF4E mRNA export pathway and impairs eIF4E-dependent oncogenic transformation. eIF4E overcomes these inhibitory mechanisms by indirectly lowering RANBP2 levels. Thus, the reprogramming of mRNA export allows this oncogene to control cell proliferation [355], as also demonstrated in acute myeloid leukemia (AML). The nuclear accumulation of eIF4E in AML patients correlates with an increase in the eIF4E-dependent export of oncoprotein-encoding transcripts. Importin-8 is involved in the direct import of eIF4E into the nuclei. Patients with AML have high levels of importin-8, leading to the increased accumulation of eIF4E in the nucleus. Thus, the importin-8-eIF4E complex is regarded as a new target for cancer therapy [382]. The phosphorylation of nuclear eIF4E is crucial for the proper control of mRNA export and oncogenic activity. Mitogen-activated protein kinase interacting kinases (MNKs) control the phosphorylation of eIF4E, and some other signaling pathways are also involved in the control of eIF4E activity [383].

eIF5A1 accumulates at high levels in the cytoplasm and nuclei of lung tumor cells, A549 cells, and lymphocytic cells [64,384,385]. eIF5A1 expression is also altered in esophageal cancer. eIF5A1 is rapidly translocated to the nucleus by tumor necrosis factor α (TNF-α), death receptor activation, or treatment with actinomycin D in colon adenocarcinoma cells. Unhypusinated eIF5A1, which is capable of nuclear localization, has pro-apoptotic functions in the nuclear form [386].

eIF5A may participate in oncogenesis by altering nucleocytoplasmic transport [387]. High levels of eIF5A2 in the nucleus and cytoplasm lead to low survival rates among patients with melanoma. eIF5A2 is a downstream target of the PI3K/Akt pathway and may induce the epithelial–mesenchymal transition [388,389]. The increased expression of eIF5A2 is associated with metastasis, angiogenesis, and shorter survival times in patients with esophageal squamous cell carcinoma. eIF5A2 may also act via the HIF1α-mediated signaling pathway [226].

eEF1A is required for the growth of tumor cells. Various eEF1A isoforms can be found in the nuclear fractions of T-lymphoblast cancer cells. eEF1A is the main nuclear protein that specifically recognizes aptameric cytotoxic oligonucleotides in these cells. By contrast, nuclear eEF1A in normal human lymphocytes does not show such activity [390]. The oncogene *PTI-1* encodes a truncated version of eEF1A, which localizes to the nucleus [391]. The nuclear localization and interaction of eEF1A and eEF1B subunits appear to contribute to cancer development in some cases [392]. The nuclear CSK-dependent localization of eEF2 is associated with aneuploidy formation, which is directly linked to malignant transformation [148].

## 9. Nuclear Translation Hypothesis

The nuclear localization of multiple CTAs has served as the basis for the nuclear translation hypothesis. The first papers describing nuclear translation were published in the middle of the 20th century [393,394] but were not subjected to criticism at that time, as the classical paradigm of separation between transcription and translation was just emerging. In the early 2000s, a hypothesis regarding nuclear translation was proposed [395], which was met with substantial criticism [396,397].

In pioneering work [395], permeabilized HeLa cells and extracted mammalian nuclei were incubated with labeled leucine and lysine-tRNAs. After incubation, newly synthesized polypeptides were found to be associated with discrete transcription factors. The hypothesis of “proofreading” for newly synthesized transcripts at the transcription loci was proposed. This model also suggests that NMD could occur directly in the nucleus [398].

New arguments for nuclear translation continue to be introduced. The formation of mature 80S ribosomes in the nucleoplasm was described [399], and the direct visualization of nuclear translation was performed [400]. An intriguing mechanism for the synthesis of peptides presented on major histocompatibility complex (MHC) class I molecules in T cells was suggested. Peptides might be synthesized on a pre-mRNA or intron template prior to mRNA splicing during the pioneer round of translation [401,402]. Moreover, these peptides can serve as tumor-associated antigens [403]. In general, nuclear translation is expected to generate multiple, short-lived peptides [404], implying an additional important functional output for nuclear translation, which can be implemented during cancer treatment [405,406].

Noncanonical nuclear translation is thought to occur without the participation of a complete set of factors required for the cytoplasmic synthesis of full-length proteins [402]. Solid-like amyloid bodies could coordinate local nuclear protein synthesis by concentrating RPs and CTAs in the nucleus during stress [407]. Despite the use of a substantially wide range of methods to confirm the existence of nuclear translation, this model remains poorly accepted by the wider scientific community [408].

## 10. Conclusions

Multiple cytoplasmic factors, referred to as CTAs, have been found in the nuclei of various cell types, where they demonstrate a wide variety of functions. An overview of the principal findings in this field is provided in Table 1 and Figure 1. The data collected to date indicate that most CTAs commonly demonstrate a dual nuclear and cytoplasmic localization; however, different sets of CTAs present with nuclear localization in different cells and under different conditions.

The functional significance of some of CTAs in nuclei has been studied in detail, whereas the roles of others remain to be clarified. Despite a significant amount of experimental data in this field, general concepts of CTA functioning in the cell nucleus are poorly developed. The mechanisms through which translation factors are imported into the nucleus are unknown for many factors. Our analysis of predicted NLS patterns in core translation factors expressed in humans (NLSdb tool [409] was used) did not identify an NLS in most cases, which may be due to the poor prediction abilities of this instrument. Another possibility involves the joint import of translation factors with specific partners, as described for eIF4E.

Whether CTAs in the nucleus function as independent proteins or as components of larger complexes requires further study; for example, the subunits of eIF3 display various nuclear functions, but their cooperation in the nucleus remains unclear. A hypothesis regarding differences in the nuclear and cytoplasmic eIF3 complex variants was suggested [224].

One obvious functional output of the nuclear localization of specific CTAs is the prevention of interactions with cytosolic CTAs, disrupting cellular translation, as was demonstrated for PABPC. However, in most cases, nuclear CTAs appear to possess specific nuclear functions.

CTAs in the nucleus are involved in multiple processes, including genome integrity control, DNA repair, replication, and nuclear stages of gene expression. Participation in gene expression appears to be the predominant function, including the regulation of TF DNA binding, TF activity and stability, and mRNA synthesis, processing, splicing, and export. An association between CTAs and silent chromatin was also shown. At the molecular level, CTAs bind proteins and RNAs, control their intranuclear localization, and modulate their interactions and activities.

CTAs contribute to the formation and function of some subnuclear structures, particularly the nucleolus, which plays an important role in the control of protein trafficking and many forms of the stress response [410]. Biochemical purifications showed an association between CTAs and architectural nuclear structures, indicating a putative structural function for CTAs (particularly RPs).

The nuclear localization of CTAs in many cases is regulated by external stimuli and internal pathways, which likely accounts for discrepancies observed in nuclear localization data reported by different studies, as the nuclear entry of a CTA may be induced by different signals in different cell types. In this model, CTAs may play optional or compensatory roles in various nuclear processes. This mode of function is confirmed by the existence of exclusively nuclear CTAs paralogs (eIF1AD, eIF4A3, eEF1BδL, and PTI-1), which share the domains, involved in some nuclear processes. The auxiliary functions of CTAs are hijacked by viruses to run their own cellular programs. RPs are common viral targets during cellular infections [411].

CTAs are abundant cell proteins and represent the targets of various signaling pathways in the cytoplasm [412]. Thus, CTAs appear to serve as reliable tools for the delivery of certain messages to the nucleus, where they can modulate nuclear processes. RPs have been suggested to play a substantial role in the cell stress response due to extraribosomal functions [186,413]. Moreover, RPs are involved in the realization of specific gene expression patterns [414]. The integration of general CTAs into cell response pathways appears to be intrinsically related to their roles in cancer phenotype development. The misregulation of certain cellular cascades leads to oncogenesis, which is partially realized by the modulation of the nuclear functions of CTAs.

Some nuclear activities of CTAs are evidently related to their ability to bind or participate in RNA metabolism. Growing evidence has indicated the crucial role of RNA and RNA-binding proteins in nuclear structure and function [33,287,415,416,417]. RNA-binding proteins also contribute significantly to eukaryotic cell regulatory networks, which are particularly important for more complex eukaryotes [418].

In addition to CTAs with additional nuclear functions, as described here, various multifunctional factors can affect the steps of the gene expression process, many of which are RNA-binding factors. For example, multifunctional factor Ago2 is involved in translational regulation and multiple nuclear activities, including chromatin regulation, transcription repression and activation, splicing, and DNA repair [419]. The Y-box binding protein (YB1) protein regulates translation but is also involved in transcription, splicing, and DNA repair [420,421,422].

CTAs could be regarded as multifunctional or moonlighting proteins. Moonlighting proteins were initially proposed to coordinate several cellular activities and participate in the cellular response to environmental signals. Switching between functions occurs due to the binding of specific molecules, other protein partners, or modifications [423]. These features can clearly be observed for CTAs. Similar to other moonlighting proteins, most CTAs contribute significantly to the complexity of cellular metabolism and the cell stress response [424]. Finally, moonlighting proteins are often associated with diseases, especially cancer development [425,426]. An analysis of CTA functions strongly supports a view that virtually no protein in the cell has only one specific function, and almost all proteins can be regarded as multifunctional [427].

The relationship between transcription and translation in eukaryotes remains poorly understood [428]; however, a deep interconnection between these stages of protein expression is expected. An integrative response to stress at all levels of the gene expression process was recently described for *Arabidopsis* [429]. A systematic analysis of protein–protein interactions predicted the extensive coupling of gene expression subprocesses, including translation, transcription, and mRNA metabolism [430]. CTAs and other multifunctional factors might serve as components of cellular mechanisms that couple the various stages of a global gene expression process. However, to date, the effects of specific CTAs at all stages of the gene expression process and the global interplay between CTAs have very rarely been investigated. Factors with integral effects on gene expression at various levels represent promising targets for applied research, including the development of treatments for viral infections and tumors.

In conclusion, the nuclear localization of CTAs is a common phenomenon and often serves as a cellular response to stress conditions and specific stimuli. Additional systematic and complex studies at all stages of gene expression examining the participation of multiple proteins and RNAs remain necessary to decipher this layer of the integrative cell response.

## Figures and Tables

**Figure 1 cells-10-03239-f001:**
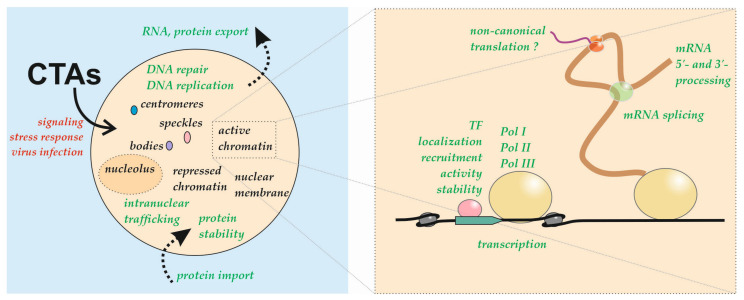
Components of the translation apparatus (CTAs) relocalize to the nucleus under various conditions (red). The intranuclear localization (black) and nuclear processes (green) in which CTAs participate are indicated. TF, transcription factor; Pol, RNA polymerase.

**Table 1 cells-10-03239-t001:** Overview of the nuclear localization and functions of core components of the translation apparatus (CTAs).

Protein (RP Names are Indicated According to [3])	Nuclear Functions (Pooled Data for All Species)	Species in which Nuclear Localization Was Detected	References
RPSA	Controls RNF8 and BRCA1 trafficking Localized at active loci on chromosomes Chromatin-associated Found in the insoluble nuclear fraction	Mammals, flies, and plants	[25,26,33,42,129,149,150]
RPS2	Binds Mdm2 and ZNF277 Localized at active loci on chromosomes Associated with HP1a Associated with RNAP II Associated with histone H1 Chromatin-associated Component of interchromatin granule clusters Found in the insoluble nuclear fraction Found in nuclei of sperm	Mammals and flies	[33,40,42,149,150,157,158,160,170,188,189,216]
RPS3	Component of DNA damage response pathway Endonuclease activity Interacts with OGG1, APE/Ref-1, UNG, TFIIH, and RECQL4 repair proteins Induction of proapoptotic genes Essential subunit of NF-κB Binds Mdm2, p53, E2F1, and ERH Associated with HP1a Associated with RNAP II Chromatin-associated Found in the insoluble nuclear fraction	Mammals, flies, and plants	[25,26,33,42,54,96,108,114,115,116,117,118,119,120,121,122,123,124,125,126,127,128,134,158,179,181,182,183,184,185,186,216,294]
RPS3A	Binds CHOP, EBNA5, and PARP Interacts with the U5 and U11 snRNAs Binds phosphatidylinositol trisphosphate Associated with HP1a Associated with RNAP II Associated with histone H1 Chromatin-associated Component of interchromatin granule clusters Found in the chromosome scaffold fraction Found in the insoluble nuclear fraction Found in nuclei of sperm	Mammals, flies, and plants	[25,26,33,40,41,42,107,157,158,160,167,168,169,216,326]
RPS4	Localized at active loci on chromosomes Associated with TFIIIE Associated with HP1a Associated with RNAP II Chromatin-associated Component of interchromatin granule clusters Found in the chromosome scaffold fraction Found in the insoluble nuclear fraction	Mammals, flies, yeasts, and plants	[25,26,33,40,41,42,149,150,158,161,216]
RPS5	Localized at active loci on chromosomes Associated with TFIIIE Associated with HP1a Associated with RNAP II Chromatin-associated Component of interchromatin granule clusters Found in the insoluble nuclear fraction	Mammals, flies, yeasts, and plants	[25,26,33,40,42,149,150,158,161,216]
RPS6	Binds and downregulates promoters of rRNA genes Interacts with NAP1 Interacts with LANA Associated with HP1a Associated with RNAP II Chromatin-associated Component of interchromatin granule clusters Found in the chromosome scaffold fraction Found in the insoluble nuclear fraction Found in nuclei of sperm	Mammals and plants	[25,26,33,40,41,42,111,158,160,212,213,216]
RPS7	Binds Mdm2, GADD45α, and BCCIPβ Associated with nascent mRNA Chromatin-associated Component of interchromatin granule clusters Found in the insoluble nuclear fraction Found in nuclei of sperm	Mammals, yeasts, and plants	[25,26,33,40,42,105,152,160,186,191,208]
RPS7A	Unknown	Plants	[26]
RPS8	Localized at active loci on chromosomes Associated with TFIIIE Associated with HP1a Associated with RNAP II Associated with histone H1 Chromatin-associated Component of interchromatin granule clusters Found in the insoluble nuclear fraction Found in the chromosome scaffold fraction	Mammals, flies, yeasts, and plants	[25,26,33,40,41,42,149,150,157,158,161,216]
RPS9	Localized at active loci on chromosomes Suppression of splicing of its own pre-mRNA Interacts with nucleophosmin Associated with TFIIIE Associated with HP1a Associated with RNAP II Associated with histone H1 Chromatin-associated Component of interchromatin granule clusters Found in the insoluble nuclear fraction Found in nuclei of sperm	Mammals, flies, and yeasts	[33,40,42,103,104,149,150,157,158,160,161,204,216,306]
RPS10	Associated with HP1a Chromatin-associated Component of interchromatin granule clusters Found in the chromosome scaffold fraction Found in the insoluble nuclear fraction	Mammals, flies, and plants	[25,26,33,40,41,42,95,158]
RPS11	Localized at active loci on chromosomes Associated with HP1a Associated with RNAP II Chromatin-associated	Mammals, flies, and plants	[25,26,33,149,150,158,216]
RPS12	Localized at active loci on chromosomes Associated with RNAP II Chromatin-associated	Mammals, flies, and plants	[25,26,33,149,150,216]
RPS13	Suppression of splicing of its own pre-mRNA Localized at active loci on chromosomes Associated with HP1a Associated with RNAP II Associated with histone H1 Chromatin-associated Found in the insoluble nuclear fraction	Mammals, flies, and plants	[25,26,33,42,149,150,157,158,216,305]
RPS14	Recruited to DNA damage sites Binds Mdm2 and c-myc Localized at active loci on chromosomes Inhibits transcription of its own gene Suppression of splicing of its own pre-mRNA Associated with HP1a Associated with RNAP II Chromatin-associated Component of interchromatin granule clusters Found in the insoluble nuclear fraction	Mammals, flies, and yeasts	[33,40,42,130,149,150,153,158,165,186,216,307,308]
RPS15	Binds Mdm2 Localized at active loci on chromosomes Associated with HP1a Associated with RNAP II Associated with histone H1 Chromatin-associated Found in the insoluble nuclear fraction	Mammals, flies, and plants	[25,26,33,42,149,150,157,158,186,216]
RPS15A	Associated with RNAP II	Humans and plants	[25,26,216]
RPS16	Associated with HP1a Associated with RNAP II Component of interchromatin granule clusters Found in the insoluble nuclear fraction	Mammals, flies, and plants	[26,40,42,158,216]
RPS17	Associated with HP1a Found in the insoluble nuclear fraction	Humans, flies, and plants	[25,26,42,158]
RPS18	Localized at active loci on chromosomes Associated with HP1a Associated with RNAP II Chromatin-associated Found in the insoluble nuclear fraction	Mammals, flies, and plants	[26,33,42,149,150,158,216]
RPS19	Associated with HP1a Found in the insoluble nuclear fraction	Humans, flies, and plants	[25,26,42,158]
RPS20	Binds Mdm2 Involved in RNAP III transcription Associated with HP1a Associated with RNAP II Chromatin-associated Found in the insoluble nuclear fraction	Mammals, flies, and yeasts	[33,42,158,163,186,216]
RPS21	Chromatin-associated Found in the insoluble nuclear fraction	Mammals and plants	[26,33,42]
RPS23	Associated with HP1a Associated with histone H1 Chromatin-associated	Mammals, flies, and plants	[25,33,157,158]
RPS24	Localized at active loci on chromosomes Associated with TFIIIE Associated with HP1a Associated with RNAP II Associated with histone H1 Chromatin-associated Found in the insoluble nuclear fraction	Mammals, flies, and plants	[25,26,33,42,149,150,157,158,216]
RPS25	Binds Mdm2 Associated with TFIIIE Associated with HP1a Associated with RNAP II Chromatin-associated Found in the insoluble nuclear fraction Found in nuclei of sperm	Mammals, flies, yeasts, and plants	[25,26,33,42,158,160,161,186,216]
RPS26	Interacts with Mdm2, p53, and p300 Suppression of splicing of its own pre-mRNA Associated with HP1a Associated with RNAP II Chromatin-associated	Mammals, flies, and plants	[25,26,33,158,186,202,216,309]
RPS27	Binds FANCD2 and FANCI Binds Mdm2 Associated with RNAP II Chromatin-associated	Mammals and plants	[26,33,131,186,216]
RPS27A	Binds Mdm2 Chromatin-associated	Mammals and plants	[25,33,192,205]
RPS28	Chromatin-associated Associated with RNAP II Component of interchromatin granule clusters	Mammals and plants	[26,33,40,216]
RPS29	Associated with HP1a	Flies	[158]
RPS30	Localized at active loci on chromosomes	Flies and plants	[26,149,150]
RPLP0	Interacts with APE1/Ref-1 Endonuclease activity Modifies position effect variegation Associated with HP1a Associated with RNAP II Associated with histone H1 Chromatin-associated Component of interchromatin granule clusters Found in the insoluble nuclear fraction	Mammals and flies	[33,40,42,54,132,133,157,158,211,216]
RPLP1	Intrinsic transactivation potential Associated with HP1a Chromatin-associated Component of interchromatin granule clusters	Mammals, flies, yeasts, and plants	[25,26,33,40,158,171]
RPLP2	Intrinsic transactivation potential Associated with HP1a Chromatin-associated Found in the insoluble nuclear fraction	Mammals, flies, yeasts, and plants	[25,26,33,42,158,171]
RPL3	Control of DNA repair Binds Sp1 Downregulates E2F1-dependent promoters by sequestering PARP1 Controls alternative splicing of its own pre-mRNA Associated with HP1a Associated with RNAP II Associated with histone H1 Chromatin-associated Found in the chromosome scaffold fraction Found in the insoluble nuclear fraction Found in nuclei of sperm	Mammals, flies, and plants	[26,33,41,42,97,135,157,158,160,166,206,216,310,321,322]
RPL4	Suppression of splicing of its own pre-mRNA Binds Mdm2, EBNA1 Associated with HP1a Associated with RNAP II Associated with histone H1 Chromatin-associated Component of interchromatin granule clusters Found in the chromosome scaffold fraction Found in the insoluble nuclear fraction	Mammals, frogs, flies, and plants	[26,33,40,41,42,142,157,158,187,216,311]
RPL5	Determines nucleolar localization of the NVL2 Interacts with splicing factor SRSF1 Binds Mdm2 and SPIN1 Induces transcriptional activity of TAp73 Participates in telomere length set point Associated with HP1a Associated with RNAP II Associated with histone H1 Chromatin-associated Found in the insoluble nuclear fraction Localized at active loci on chromosomes	Mammals, flies, and plants	[25,26,33,42,87,88,140,149,150,157,158,166,186,195,201,207,216,375]
RPL6	Interacts with histone H2A/H2AX Recruited to DNA damage sites, mediates recruitment of repair proteins Binds Mdm2 Mediates DNA binding of Tax Associated with TFIIIE Associated with HP1a Associated with RNAP II Associated with histone H1 Chromatin-associated Found in the insoluble nuclear fraction	Mammals, flies, yeasts, and plants	[25,26,33,42,130,157,158,161,175,190,216]
RPL7	Associated with centromeres Localized at active loci on chromosomes Counteracts the binding of VDR-RXR to DNA Associated with nascent mRNA Interacts with histone H1 Associated with TFIIIE Associated with HP1a Associated with RNAP II Associated with histone H1 Chromatin-associated Found in the insoluble nuclear fraction	Mammals, flies, yeasts, and plants	[26,33,42,149,150,151,152,155,157,158,161,176,216]
RPL7A	Localized at active loci on chromosomes Associated with HP1a Associated with RNAP II Associated with histone H1 Chromatin-associated Component of interchromatin granule clusters Found in the insoluble nuclear fraction	Mammals, flies, and plants	[26,33,40,42,149,150,157,158,216]
RPL8	Recruited to DNA damage sites Localized at active loci on chromosomes Associated with HP1a Associated with RNAP II Associated with histone H1 Chromatin-associated Component of interchromatin granule clusters	Mammals and flies	[33,40,130,149,150,157,158,216]
RPL9	Participates in intranuclear traffic of Gag protein Associated with HP1a Associated with RNAP II Chromatin-associated Found in the insoluble nuclear fraction Found in nuclei of sperm	Mammals, flies, and plants	[26,33,42,102,109,158,160,216]
RPL10	Suppression of the binding of c-Jun homodimer to DNA Associated with RNAP II Associated with histone H1 Chromatin-associated Found in the insoluble nuclear fraction	Mammals and plants	[25,26,33,42,94,95,157,178,216]
RPL10A	Interacts with LIMYB and downregulates expression of RP genes Suppression of splicing of its own pre-mRNA Associated with HP1a Associated with RNAP II Found in the insoluble nuclear fraction	Humans, flies, and plants	[42,158,173,174,216,312]
RPL11	Binds c-myc, Mdm2, ARF, GRWD1, PML, and PICT1 Induces transcriptional activity of TAp73 Counteracts binding of PPARα DNA Stimulates p53-mediated transcription Induces apoptosis Nucleolar localization promotes tumor progression Association with centromeres Localized at active loci on chromosomes Required for integrity of the nucleolar structure Associated with HP1a Associated with RNAP II Chromatin-associated Found in the insoluble nuclear fraction	Mammals, flies, yeasts, and plants	[25,26,33,42,84,110,149,150,151,158,162,164,166,177,186,195,196,199,200,207,216,372,373,374]
RPL12	Localized at active loci on chromosomes Interacts with Corto Required for transcription of the PHO pathway genes Suppression of splicing of its own pre-mRNA Associated with HP1a Associated with histone H1 Chromatin-associated Component of interchromatin granule clusters Found in the insoluble nuclear fraction	Mammals, flies, worms, yeasts, and plants	[26,33,40,42,149,150,154,157,158,209,210,313]
RPL13	Promotes activity of *NF-κB* and *IFN-β* promoters Associated with TFIIIE Associated with HP1a Associated with RNAP II Chromatin-associated Component of interchromatin granule clusters Found in the insoluble nuclear fraction	Mammals, flies, yeasts, and plants	[25,26,33,40,42,158,161,172,216]
RPL13A	Associated with HP1a Associated with RNAP II Found in the insoluble nuclear fraction	Humans and flies	[42,158,216]
RPL14	Localized at active loci on chromosomes Associated with TFIIIE Associated with HP1a Associated with RNAP II Chromatin-associated Component of interchromatin granule clusters Found in the insoluble nuclear fraction	Mammals, flies, yeasts, and plants	[25,26,33,40,42,149,150,158,161,216]
RPL15	Required for integrity of the nucleolar structure Localized at active loci on chromosomes Associated with HP1a Associated with RNAP II Chromatin-associated Component of interchromatin granule clusters Found in the insoluble nuclear fraction	Mammals, flies, and plants	[26,33,40,42,84,149,150,158,216]
RPL17	Localized at active loci on chromosomes Associated with HP1a Associated with histone H1 Chromatin-associated Found in the insoluble nuclear fraction	Mammals, flies, and plants	[26,33,42,149,150,157,158]
RPL18	Localized at active loci on chromosomes Associated with HP1a Associated with histone H1 Chromatin-associated Found in the insoluble nuclear fraction	Mammals, flies, and plants	[25,26,33,42,149,150,157,158]
RPL18A	Associated with HP1a Associated with RNAP II Found in the insoluble nuclear fraction	Humans, flies, and plants	[25,26,42,158,216]
RPL19	Interacts with ERH Associated with HP1a Associated with RNAP II Chromatin-associated Component of interchromatin granule clusters Found in the insoluble nuclear fraction	Mammals, flies, and plants	[25,26,33,40,42,108,158,216]
RPL21	Associated with HP1a Associated with RNAP II Associated with histone H1 Chromatin-associated Found in the insoluble nuclear fraction	Mammals, flies, and plants	[25,26,33,42,157,158,216]
RPL22	Binds Mdm2 Participates in alternative splicing of *smad2* pre-mRNA Suppression of splicing of paralog pre-mRNA Interacts with histone H1 Associated with HP1a Associated with RNAP II Associated with histone H1 Chromatin-associated	Mammals, fish, flies, and plants	[25,26,33,98,99,100,155,157,158,193,216,314,323,324]
RPL23	Binds Mdm2 and nucleophosmin Associated with HP1a Chromatin-associated Component of interchromatin granule clusters Found in the insoluble nuclear fraction	Humans, flies, and plants	[25,26,33,40,42,106,158,186,242]
RPL23A	Localized at active loci on chromosomes Associated with HP1a Associated with RNAP II Chromatin-associated Component of interchromatin granule clusters Found in the insoluble nuclear fraction	Mammals, flies, and plants	[25,33,40,42,151,158,216]
RPL24	Binds pre-miRNA and facilitates its processing Associated with HP1a Associated with RNAP II Chromatin-associated Found in the insoluble nuclear fraction	Mammals, flies, and plants	[25,26,33,42,158,216,327]
RPL26	Participates in alternative splicing of *p53* pre-mRNA Binds Mdm2 Associated with nascent mRNA Associated with TFIIIE Associated with HP1a Chromatin-associated Found in the insoluble nuclear fraction	Mammals, flies, and yeasts	[33,42,152,158,161,186,325]
RPL27	Associated with HP1a Chromatin-associated Found in the insoluble nuclear fraction	Mammals, flies, and plants	[25,26,33,42,158]
RPL27A	Associated with HP1a Associated with histone H1 Chromatin-associated Component of interchromatin granule clusters Found in the insoluble nuclear fraction	Mammals and flies	[33,40,42,157,158]
RPL28	Localized at active loci on chromosomes Associated with HP1a Associated with RNAP II Chromatin-associated	Mammals, flies, and plants	[25,26,33,88,149,150,158,216]
RPL29	Associated with HP1a Associated with RNAP II	Humans, flies, and plants	[25,26,158,216]
RPL30	Suppression of splicing of its own pre-mRNA Associated with HP1a Associated with RNAP II Associated with histone H1 Chromatin-associated	Mammals, flies, yeasts, and plants	[25,26,33,157,158,216,315,316]
RPL31	Associated with HP1a Associated with RNAP II Chromatin-associated Component of interchromatin granule clusters Found in the insoluble nuclear fraction	Mammals, flies, and plants	[26,33,40,42,158,216]
RPL32	Suppression of splicing of its own pre-mRNA Localized at active loci on chromosomes Associated with HP1a Associated with RNAP II Chromatin-associated Found in the insoluble nuclear fraction	Mammals, flies, and plants	[25,26,33,42,149,150,158,216,317,318]
RPL34	Localized at active loci on chromosomes Associated with nascent mRNA Associated with HP1a Associated with RNAP II	Humans, flies, yeasts, and plants	[25,26,149,150,152,158,216]
RPL35	Associated with HP1a Associated with RNAP II Chromatin-associated Component of interchromatin granule clusters Found in the insoluble nuclear fraction	Mammals, flies, and plants	[26,33,40,42,158,216]
RPL35A	Associated with HP1a Associated with RNAP II Chromatin-associated	Mammals, flies, and plants	[26,33,158,216]
RPL36	Localized at active loci on chromosomes Associated with TFIIIE Associated with HP1a Associated with RNAP II Chromatin-associated	Mammals, flies, yeasts, and plants	[25,26,33,149,150,158,161,216]
RPL36A	Chromatin-associated	Mammals	[33]
RPL37	Binds Mdm2	Humans	[186,203]
RPL37A	Associated with HP1a Chromatin-associated	Mammals and flies	[33,158]
RPL38	Chromatin-associated	Mammals and plants	[26,33]
RPL39	Localized at active loci on chromosomes	Flies and plants	[26,149,150]
RPL41	Induces export of ATF4 from nuclei	Humans	[246]
eIF1	Chromatin-associated	Mammals	[27,33]
eIF1A	Chromatin-associated	Mammals, flies, and yeasts	[20,22,27,28,33]
eIF2 (S1–S3 subunits)	Stabilization of DNA-PKcs-Ku complex Controls RRN3/TIF-IA and RNAP I activity Associated with HP1a Associated with RNAP II Chromatin-associated	Mammals, flies, yeasts, and plants	[20,23,25,26,33,50,51,52,53,136,158,216,220]
eIF2A	Associated with mRNA 3′-processing complex Chromatin-associated	Mammals	[33,332]
eIF2B (1–5 subunits)	Chromatin-associated	Mammals and yeasts	[20,33,54]
eIF3a	Interacts with RNAP II Associated with mRNA 3′-processing complex Associated with histone H1 Chromatin-associated	Mammals and plants	[26,33,157,214,332]
eIF3b	Associated with mRNA 3′-processing complex Associated with histone H1 Chromatin-associated	Mammals and plants	[26,33,54,157,332]
eIF3c	Interacts with RNAP II Interacts with the nuclear COP9 signalosome Chromatin-associated	Mammals, yeasts, and plants	[26,33,49,54,215]
eIF3d	Associated with HP1a Chromatin-associated	Mammals, flies, yeasts, and plants	[20,26,33,158]
eIF3e	Participates in DNA repair Interacts with ATM Promotes loading of the RAD51 Interacts with Ub-MCM7 Interacts with the nuclear COP9 signalosomeProper accumulation of proteasome in the nucleus Interacts with Rfp, HIF-2α Increases the transcriptional activity of Pap1 Colocalized in certain PML nuclear bodies Copurifies with CBP80 Associated with mRNA 3′-processing complex Chromatin-associated	Mammals and plants	[26,33,38,45,46,47,48,70,137,138,141,221,249,332,335,379]
eIF3f	Cooperates with STAT3 and other TFs Associated with mRNA 3′-processing complex Colocalized in nucleus with CDK11 Chromatin-associated	Mammals and plants	[24,25,26,33,157,222,223,224,332]
eIF3g	Chromatin-associated	Mammals, yeasts, and plants	[20,25,26,33,54]
eIF3h	Enhancer of variegation function Associated with mRNA 3′-processing complex Chromatin-associated	Mammals and plants	[26,33,225,332]
eIF3i	Interacts with RNAP II Associated with mRNA 3′-processing complex Associated with histone H1 Chromatin-associated	Mammals and yeasts	[20,33,157,214,332]
eIF3j	Interacts with RNAP II	Yeasts	[215]
eIF3k	Colocalized with PML bodies Interacts with cyclin D3	Humans and plants	[26,36,37]
eIF3l	Cofactor of RNAP I Associated with histone H1 Chromatin-associated	Mammals	[33,157,218,219]
eIF3m	Interacts with RNAP II Chromatin-associated	Mammals	[33,214]
eIF4E	Defines specific pathway of mRNA export Binds LRPPRC Interacts with PML Interacts with factors of mRNA 3′-processing Participates in mRNA biogenesis Stimulates mRNA capping and alternative splicing Associated with splicing factors Participates in cancer cell transformation	Humans, flies, frogs, yeasts, and plants	[17,18,19,24,25,26,32,39,65,66,329,330,331,354,355,356,357,358,359,380,381,382,383]
eIF4A	Associated with mRNA 3′-processing complex Interacts with CBP80 Presumable role in splicing Associated with HP1a and HP1c Associated with RNAP II Chromatin-associated Component of interchromatin granule clusters Found in the insoluble nuclear fraction	Mammals, flies, and plants	[25,26,33,34,40,42,71,158,159,216,332,336,337]
eIF4G	Associated with mRNA 3′-processing complex Interacts with the nuclear CBC and splicing factors Localized in XY body of spermatocytes Associated with HP1a Associated with RNAP II Chromatin-associated	Mammals, flies, and plants	[25,26,33,54,72,158,216,332,342,343,344]
eIF4B	Associated with HP1a Chromatin-associated	Mammals, flies, and plants	[26,33,54,158]
eIF4H	Chromatin-associated	Mammals	[33]
eIF5	Chromatin-associated	Mammals and plants	[24,33]
eIF5B	Active transcription sites Associated with RNAP II Associated with histone H1	Humans, flies, and plants	[26,54,150,157,216]
eIF6	Participates in ribosome biogenesis Associated with RNAP II Chromatin-associated Found in the insoluble nuclear fraction	Mammals, frogs, yeasts, and plants	[20,21,25,26,27,30,32,33,34,42,106,216]
eEF1A	Stimulates the recruitment of HSF1 to DNA and activity of *HSP70* Recruited to *IFN-γ* promoter and stimulates transcription Interacts with ZPR1, RNAP II, Zw5, ZIPIC, and Grau Promotes nuclear export of Sox10 and Snail Stabilizes RNAP II-TAR RNA interaction Associated with mRNA 3′-processing complex Defines a specific protein export pathway mRNA export, tRNA re-export Associated with HP1a Chromatin-associated Found in the insoluble nuclear fraction	Humans, flies, yeasts, and plants	[20,24,27,33,34,42,158,216,227,228,229,230,231,232,233,234,235,253,254,255,256,257,258,332,333,334,390,391,392]
eEF1Bβ	Chromatin-associated Putative role in transcription, splicing, and DNA damage response	Mammals and plants	[25,26,33,236,392]
eEF1Bγ	Putative role in splicing and control of mRNA stability Binds RNAP II and recruited to several promoters Associated with mRNA 3′-processing complex Chromatin-associated Found in the insoluble nuclear fraction	Mammals, flies, and plants	[25,26,33,42,236,237,238,239,332,392]
eEF1Bδ	Chromatin-associated	Mammals and plants	[26,33]
eEF2	Induces nuclear morphological changes and aneuploidy Associated with HP1a Chromatin-associated Found in the insoluble nuclear fraction	Mammals, flies, and plants	[25,26,33,42,148,158]
eIF5A	Induction of apoptosis Associated with intranuclear filaments of NPC Binds the promoter of the *HIF1α* and activates its transcription Part of a specific RNA nuclear export pathway Associated with oncogenesis Associated with HP1a Chromatin-associated Found in the insoluble nuclear fraction	Mammals, flies, yeasts, and plants	[20,25,26,33,42,55,56,57,58,59,60,61,62,63,64,158,226,361,362,363,364,365,366,367,368,369,384,385,386,387,388,389]
eEFSec	Unknown	Humans and frogs	[31,32]
eRF1	The quality control mechanism of maturing ribosomes Chromatin-associated	Mammals, yeasts, and plants	[20,25,35]
eRF3	Found at active transcription sites Chromatin-associated	Mammals and flies	[150]

## Data Availability

Data sharing not applicable.

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
