# Peer review of "Localization and Functional Roles of Components of the Translation Apparatus in the Eukaryotic Cell Nucleus"

_cells, 2021, doi:10.3390/cells10113239_

Round 1

Reviewer 1 Report

The manuscript by Kachaev et al., is an extensive review of the literature that has described the presence of translation factors or components of the translation machinery in the nucleus, and of the cases where they have been associated with nuclear functions. It is very comprehensive and ends with the very controversial description by some studies of nuclear translation.

This review is not the kind of review that one reads to learn something about a given topic, in fact it is quite painful to read because it is mainly an enumeration of description of findings. However, it is a very useful review that scientists can access to find out what has been published in this field and find the link to the references which describe the work. Because it is also well written and well organized I do consider it as a useful review.

Author Response

We are grateful to the reviewer for the positive evaluation of our manuscript.

We have rearranged the material to provide a more structured view. Multiple issues throughout the text and table have been fixed.

A detailed reply is given below.

The manuscript by Kachaev et al., is an extensive review of the literature that has described the presence of translation factors or components of the translation machinery in the nucleus, and of the cases where they have been associated with nuclear functions. It is very comprehensive and ends with the very controversial description by some studies of nuclear translation.

As studies of nuclear translation are published permanently, we have collected the relevant papers and provided an overview of this topic. Poor acceptance of this hypothesis is indicated (line 750).

This review is not the kind of review that one reads to learn something about a given topic, in fact it is quite painful to read because it is mainly an enumeration of description of findings. However, it is a very useful review that scientists can access to find out what has been published in this field and find the link to the references which describe the work. Because it is also well written and well organized I do consider it as a useful review.

We agree with the notion. Our paper reflects the current state of this field. The latter is overloaded with data, while lacking significant conceptualization. We provide our attempt to summarize the relevant information.

Reviewer 2 Report

In this review entitled ‘Localization and functional roles of components of the translation apparatus in the eukaryotic cell nucleus’ by Kachaev et al. author describes the nuclear roles of various components of the translation apparatus. To start with they give the overview of nuclear localization of different components of translation and then describe the individual roles of various translation factors in nuclear processes like DNA damage, transcription, mRNA processing, and export by citing appropriate references.

Overall, this review was thorough, compiled, and written well, therefore I do not have any major concerns/comments. I appreciate the authors' valiant effort to consolidate and describe the nuclear roles of translation components in various eukaryotic species, this compilation will benefit the broad readers.

I just have few minor comments to improve the manuscript quality further.

Minor comments:

1. Both the figure and table are placed in the end after the text, perhaps they should be placed better somewhere in the middle to benefit the readers more.

2. Authors use the terms ubiquitylation/ubiquitination in the mixture..while both convey a similar meaning..consistency in usage could be better.

3. Check for typos and minor grammar.

For eg: in line 130 `Arabodopsis’ to be corrected as ‘Arabidopsis

Line 476 CBp80 to be corrected as CBP80

Author Response

We are grateful to the reviewer for the positive evaluation of our manuscript.

We have rearranged the material to provide a more structured view. Multiple issues throughout the text and table have been fixed.

A detailed reply is given below.

  1. Both the figure and table are placed in the end after the text, perhaps they should be placed better somewhere in the middle to benefit the readers more.

The table 1 is quite long. In our opinion, disruption of the main text with the table would be quite inconvenient. Moreover, the table summarizes the data and does not supplement the text with additional information, but rather provides an overview of the topic.

  1. Authors use the terms ubiquitylation/ubiquitination in the mixture..while both convey a similar meaning..consistency in usage could be better.

Corrected

  1. Check for typos and minor grammar. For eg: in line 130 `Arabodopsis’ to be corrected as ‘Arabidopsis Line 476 CBp80 to be corrected as CBP80

Corrected

Reviewer 3 Report

The manuscript entitled "Localization and Functional Roles of Components of the Translation Apparatus in the Eukaryotic Cell Nucleus" by Kachaev et al. represents a fairly valuable and comprehensive review article. It is devoted to a modern and very interesting topic, the nuclear functions of translation machinery components, that undoubtedly deserves the attention of Cells readers.

Yet the review is written in a somewhat unusual format. The core of the paper is Table 1, which is very valuable and informative. But the rest of the text majorly replicates this myriad of facts – albeit in a more expanded form, yet usually with just one sentence per fact followed by a referenced source. Thus, the review is most like a list of facts, or a kind of reference book, supplemented by minimalistic comments. This is both a strong and a weak side of the review. The strong point is really VERY wide field coverage. It includes 427 (!) references, making it the most comprehensive review I have ever seen in this area. The weak one is that the facts are often poorly linked to each other, so no general concepts or new ideas are discussed throughout the text, except the last section (which draws some general conclusions) and probably the section 8 (written in a distinct, more traditional style). It is therefore not surprising that the review contains only one figure, which is probably Graphical Abstract.

Despite its unconventional style, the review can be very useful for researchers who study both canonical and moonlighting activities of the translation machinery components. Thus, I would like to recommend it to be published in Cells, once the following minor corrections are made by the authors (see below). However, I suggest the authors adding some discussion of the listed facts here and there throughout the text, as well as link them with each using more clearly expressed logics.

Another general comment is that the authors should utilize the commonly used names of translation machinery components and avoid inventing new ones. For example, for ribosomal proteins now there are only two commonly used nomenclatures: one is provided by GenBank (e.g. RPS5), while another is used by researchers who study ribosome structure, see PMID: 24524803 (e.g. uS7, which is the same as RPS5). I have never met the designations like “RpS5”, used by the authors, in modern literature. Please note that even in the Ribosomal Protein Genes (RPG) Database, which is cited by the authors, the “RPS5-type” of nomenclature is used. Even more strange is using “eTF1” and “eTF3” for eRF1/ETF1 and eRF3/GSPT proteins, respectively.

Minor points (with line numbers):

19: “Most components” – I suggest to avoid drawing a proportion, using “many” instead. Speaking truly, my personal opinion is that a lot of reports regarding moonlighting functions of translation machinery components are just fakes, caused by secondary effects of their depletion. Even if not so, the proportion is hard to be estimated correctly.

22: “fundamental” - it is probably “basic” what the authors meant here (but I do not insist)

36: not all steps of ribosome biogenesis occurs in the nucleolus (although the majority of them), as a few final steps occurs in the cytoplasm.

43: eIF2D is more likely a ribosome recycling factor, also involved in reinitiation, but not in initiation per se, at least under normal conditions. Moreover, although the recycling step is mentioned, no other recycling factors are listed (ABCE1/Rli1p, MCTS1/Tma20p, DENR/Tma22p), although this is one of the four main steps of the translation cycle.

43: not every eIF3 has 13 subunits: for example, in S.cerevisiae it consists of 5 subunits with some additional loosely associated proteins; moreover, some eIF3-studying groups do not consider eIF3j as a eIF3 subunit now; so, I would write “up to 13 subunits” here, it would be correct.

 50: it is not correct that all CTAs were thought to be exclusively localized in the cytoplasm until 1990s, as nuclear localization of at least RPs was known since early times.

62: “described in a separate work” – “study” would sound better here.

75: eIF1AD is probably a ribosome biogenesis factor (PMID: 33208940), so it should probably be out of the focus of the current review.

107: eIF4D is a former name of the elongation factor eIF5A (used back in 1970s).

262 and elsewhere: “TF”: not all abbreviations are explained when used for the first time in the text.

113: I’m not sure that mRNA-binding proteins should be reviewed, as in this case several hundreds of other mRNA-bps must be added.

115: eTT1 is probably Ett1p/Nro1p, which is a substrate adaptor for prolylhydroxylase Ofd1p; together they participate in a complex regulation of nuclear import of uS12/Rps23 (PMID: 29083304) and regulate fungal hypoxic adaptation. However, I’m not sure that the Nro1p role as a prolylhydroxylase substrate adaptor and nuclear import receptor makes it a subject of this review. By the way, Refs 21 and 53 do not clearly reflect this role of Nro1p in the uS12 regulation.

171: “extracellular signal-regulated kinase (ERK1/2)1/2” – please correct.

197: the topic of MDM2-RP complexes is a large field, which is reviewed in details in the next section; so, I would add “(see below)” after “[112]”.

202: eIF3e/Int6 is a very special eIF3 subunit, as eIF3 shares it with two other structurally similar (but functionally unrelated) complexes, COP9 signalosome and 19S proteasome lid, in both of which it seems to be a “full value” integral component. So, it is clear that this particular protein has many additional functions as a component of these complexes and not as a eIF3 subunit. I guess it should be at least declared in the text (with a reference to an appropriate review).

218, 219: why “epsilon” and not “ε”? – in other cases, the authors use Greek symbols.

270: what is “intrinsic transactivation potential”? – it is unclear.

278: “RPs play a variety of other roles” – the phase sounds strange in this particular place, as many examples of “other roles” have been described before and is described after that for RPs.

287: the human protein and the corresponding gene are also called MDM2 (https://www.uniprot.org/uniprot/Q00987, https://www.ncbi.nlm.nih.gov/gene/4193)

307: “By contrast” – it is unclear, what contrasts here with the previous sentence.

315: “RpS7, an Mdm2 partner protein…” – why it is “a partner”, it should be clarified here or above; the only information mentioned above is that RPS7 is a substrate of MDM2, this is not the same.

340: “the transcriptional activity of Pap1 TF”: I’m not an expert in transcription factors, but I suggest that these proteins do not have a “transcriptional activity” (in contrast to polymerases).

359: “a tissue-specific isoform of eEF1A2” – to me, this phrase is misleading: first, as eEF1A1 and eEF1A2 are encoded by different genes, while “isoforms” usually means some forms of a product of the same gene; second. “isoform of <some protein>” means not THIS , but some OTHER protein – while the authors probably meant eEF1A2 here.

373: “An isoform of eEF1A2” – does it mean eEF1A1? - see above.

379: “promoters for Vimentin, Che-1, and p53 in human cells” – gene symbols should be indeed italized, but in this case these are not gene symbols and likely trivial names of the corresponding proteins. I would change this for “promoters of the VIM, AATF, and TP53 genes in human cells”.

379: “Nuclear eEF1Bγ” – it is likely “Nuclear localized eEF1Bγ”

385-386: please check the sentence (“in addition – in addition”); I would also note that the term “expression level” should not be applied to “transcription factors”, which are actually proteins and not genes or mRNAs (“expression” is a process of making some secondary product – i.e. mRNA or protein; proteins do not “express” anything); for proteins, just “level” or “abundance” should be used.

388: “nucleolus, which acts”: just textually, in such a composition “which” corresponds to nucleolus and not to nucleophosmin.

394, 625: "stressful conditions" – it is better to use “stress conditions” here.

394-397: I’m not sure the eIF2α/ATF4 story is relevant to the topic; if you think it is, please rephrase somehow (like “In addition to the well-known affect…”).

423-433: it is unclear (i) which virus is the authors talking about; (ii) which organism and (iii) what particular PABPC: for example, in mammals I know at least 6 PABPC proteins.

436-437: it is unclear how PABPC1 in nucleus “knows” whether a pre-mRNA is subjected to NMD (which occurs in the cytoplasm).

449-452: I would just remove the part starting with “during the early…”

476: “CBp80”

477: what is “"transcription capping checkpoint"? – it is likely “transcript capping checkpoint" or smth like that, as “transcription” cannot be “capped”.

480-485: I would specifically emphasize the situation in budding yeast, where RP genes are among ~5% of genes containing introns, as this is a general mechanism for regulation of the RP abundance in this particular organism (PMID: 22000012, PMID: 26945043, PMID: 30651641).

507: eIF4AI or eIF4A1 – both are correct, but please unify (I would like to see eIF4A1, which is the currently used gene symbol for the human protein).

511-514: I’m not sure this is relevant to the topic, yet EJC is related to NMD and NMD is related to translation... But should you consider Y14, MAGOH etc. as well in this review (they do have a role in the nucleus by definition)?

522: again, as in the case of “eEF1A2 isoform” before: the phrase “the eIF4G1 paralog” most likely means eIF4G2 and not eIF4G1.

537: “polyA sites” – it is probably “polyadenylation sites” what the authors mean.

548: “eIF4E-4ET” looks strange: it is likely eIF4E transporter, which is usually called 4E-T or eIF4ENIF.

552: “mediateD”.

559: “aninhibitor”.

565: “mG cap”

580-587, 601-605: I suggest moving these paragraphs to the appropriate section (#2).

650: please add “some” before :other signaling”.

668: “the primary nuclear protein” – what does it mean?

714: “Our analysis of predicted NLS [407] patterns” – please rephrase this sentence, as now it can be considered as the results of your own analysis described in Ref. 407 (which is not the case).

Table 1: please consider my above comment about nomenclature of proteins.

Ref.26: please remove <em>/<em>.

Hopefully, my comments will help improve the paper!

It is my usual policy to reveal my identity to the authors: Sergey Dmitriev.

Author Response

We are grateful to the reviewer for the thorough reading and positive evaluation of our manuscript.

Multiple issues throughout the text and table have been fixed.

A detailed reply is given below.

Thus, the review is most like a list of facts, or a kind of reference book, supplemented by minimalistic comments. This is both a strong and a weak side of the review. The strong point is really VERY wide field coverage. It includes 427 (!) references, making it the most comprehensive review I have ever seen in this area. The weak one is that the facts are often poorly linked to each other, so no general concepts or new ideas are discussed throughout the text, except the last section (which draws some general conclusions) and probably the section 8 (written in a distinct, more traditional style). It is therefore not surprising that the review contains only one figure, which is probably Graphical Abstract.

We agree with the notion. Our paper reflects the current state of this field. The latter is overloaded with data, while lacking significant conceptualization. We provide our attempt to summarize the relevant information.

However, I suggest the authors adding some discussion of the listed facts here and there throughout the text, as well as link them with each using more clearly expressed logics.

We have rearranged the material to provide a more structured view.

Another general comment is that the authors should utilize the commonly used names of translation machinery components and avoid inventing new ones. For example, for ribosomal proteins now there are only two commonly used nomenclatures: one is provided by GenBank (e.g. RPS5), while another is used by researchers who study ribosome structure, see PMID: 24524803 (e.g. uS7, which is the same as RPS5). I have never met the designations like “RpS5”, used by the authors, in modern literature. Please note that even in the Ribosomal Protein Genes (RPG) Database, which is cited by the authors, the “RPS5-type” of nomenclature is used. Even more strange is using “eTF1” and “eTF3” for eRF1/ETF1 and eRF3/GSPT proteins, respectively.

The names of the proteins have been corrected.

19: “Most components” – I suggest to avoid drawing a proportion, using “many” instead. Speaking truly, my personal opinion is that a lot of reports regarding moonlighting functions of translation machinery components are just fakes, caused by secondary effects of their depletion. Even if not so, the proportion is hard to be estimated correctly.

Corrected

22: “fundamental” - it is probably “basic” what the authors meant here (but I do not insist)

Corrected

36: not all steps of ribosome biogenesis occurs in the nucleolus (although the majority of them), as a few final steps occurs in the cytoplasm.

Corrected

43: eIF2D is more likely a ribosome recycling factor, also involved in reinitiation, but not in initiation per se, at least under normal conditions. Moreover, although the recycling step is mentioned, no other recycling factors are listed (ABCE1/Rli1p, MCTS1/Tma20p, DENR/Tma22p), although this is one of the four main steps of the translation cycle.

We removed mentioning of this factor.

43: not every eIF3 has 13 subunits: for example, in S.cerevisiae it consists of 5 subunits with some additional loosely associated proteins; moreover, some eIF3-studying groups do not consider eIF3j as a eIF3 subunit now; so, I would write “up to 13 subunits” here, it would be correct.

Corrected

 50: it is not correct that all CTAs were thought to be exclusively localized in the cytoplasm until 1990s, as nuclear localization of at least RPs was known since early times.

Corrected

62: “described in a separate work” – “study” would sound better here.

Corrected

75: eIF1AD is probably a ribosome biogenesis factor (PMID: 33208940), so it should probably be out of the focus of the current review.

We mention in the text several paralogs of CTAs with exclusively nuclear functions. We discuss it in the final section (line 793).

107: eIF4D is a former name of the elongation factor eIF5A (used back in 1970s).

Corrected

262 and elsewhere: “TF”: not all abbreviations are explained when used for the first time in the text.

Corrected

113: I’m not sure that mRNA-binding proteins should be reviewed, as in this case several hundreds of other mRNA-bps must be added.

We mention just few examples of specific translation regulators to underline that the raised topic is quite broad.

115: eTT1 is probably Ett1p/Nro1p, which is a substrate adaptor for prolylhydroxylase Ofd1p; together they participate in a complex regulation of nuclear import of uS12/Rps23 (PMID: 29083304) and regulate fungal hypoxic adaptation. However, I’m not sure that the Nro1p role as a prolylhydroxylase substrate adaptor and nuclear import receptor makes it a subject of this review. By the way, Refs 21 and 53 do not clearly reflect this role of Nro1p in the uS12 regulation.

We removed this piece of information.

171: “extracellular signal-regulated kinase (ERK1/2)1/2” – please correct.

Corrected

197: the topic of MDM2-RP complexes is a large field, which is reviewed in details in the next section; so, I would add “(see below)” after “[112]”.

Added

202: eIF3e/Int6 is a very special eIF3 subunit, as eIF3 shares it with two other structurally similar (but functionally unrelated) complexes, COP9 signalosome and 19S proteasome lid, in both of which it seems to be a “full value” integral component. So, it is clear that this particular protein has many additional functions as a component of these complexes and not as a eIF3 subunit. I guess it should be at least declared in the text (with a reference to an appropriate review).

Corrected

218, 219: why “epsilon” and not “ε”? – in other cases, the authors use Greek symbols.

Corrected

270: what is “intrinsic transactivation potential”? – it is unclear.

Corrected

278: “RPs play a variety of other roles” – the phase sounds strange in this particular place, as many examples of “other roles” have been described before and is described after that for RPs.

Corrected

287: the human protein and the corresponding gene are also called MDM2 (https://www.uniprot.org/uniprot/Q00987, https://www.ncbi.nlm.nih.gov/gene/4193)

Corrected

307: “By contrast” – it is unclear, what contrasts here with the previous sentence.

Corrected

315: “RpS7, an Mdm2 partner protein…” – why it is “a partner”, it should be clarified here or above; the only information mentioned above is that RPS7 is a substrate of MDM2, this is not the same.

As indicated above in the text, multiple RPs bind Mdm2 (line 331). The list of these RPs is quite long and was introduced in the Table 1.

340: “the transcriptional activity of Pap1 TF”: I’m not an expert in transcription factors, but I suggest that these proteins do not have a “transcriptional activity” (in contrast to polymerases).

Corrected

359: “a tissue-specific isoform of eEF1A2” – to me, this phrase is misleading: first, as eEF1A1 and eEF1A2 are encoded by different genes, while “isoforms” usually means some forms of a product of the same gene; second. “isoform of <some protein>” means not THIS , but some OTHER protein – while the authors probably meant eEF1A2 here.

Corrected

373: “An isoform of eEF1A2” – does it mean eEF1A1? - see above.

Corrected

379: “promoters for Vimentin, Che-1, and p53 in human cells” – gene symbols should be indeed italized, but in this case these are not gene symbols and likely trivial names of the corresponding proteins. I would change this for “promoters of the VIM, AATF, and TP53 genes in human cells”.

Corrected

379: “Nuclear eEF1Bγ” – it is likely “Nuclear localized eEF1Bγ”

Corrected

385-386: please check the sentence (“in addition – in addition”); I would also note that the term “expression level” should not be applied to “transcription factors”, which are actually proteins and not genes or mRNAs (“expression” is a process of making some secondary product – i.e. mRNA or protein; proteins do not “express” anything); for proteins, just “level” or “abundance” should be used.

Corrected

388: “nucleolus, which acts”: just textually, in such a composition “which” corresponds to nucleolus and not to nucleophosmin.

Corrected

394, 625: "stressful conditions" – it is better to use “stress conditions” here.

Corrected

394-397: I’m not sure the eIF2α/ATF4 story is relevant to the topic; if you think it is, please rephrase somehow (like “In addition to the well-known affect…”).

We describe this selective effect of a general translation factor on the level of specific TF as an example of the indirect effects of CTAs, as indicated above.

423-433: it is unclear (i) which virus is the authors talking about; (ii) which organism and (iii) what particular PABPC: for example, in mammals I know at least 6 PABPC proteins.

Corrected

436-437: it is unclear how PABPC1 in nucleus “knows” whether a pre-mRNA is subjected to NMD (which occurs in the cytoplasm).

Corrected

449-452: I would just remove the part starting with “during the early…”

Effect on early stages of transcription is different for PABPC and its partner paip2. While PABPC has no effect, paip2 participates in these stages (indicated in the text).

476: “CBp80”

Corrected

477: what is “"transcription capping checkpoint"? – it is likely “transcript capping checkpoint" or smth like that, as “transcription” cannot be “capped”.

Corrected

480-485: I would specifically emphasize the situation in budding yeast, where RP genes are among ~5% of genes containing introns, as this is a general mechanism for regulation of the RP abundance in this particular organism (PMID: 22000012, PMID: 26945043, PMID: 30651641).

This notion was added.

507: eIF4AI or eIF4A1 – both are correct, but please unify (I would like to see eIF4A1, which is the currently used gene symbol for the human protein).

Corrected

511-514: I’m not sure this is relevant to the topic, yet EJC is related to NMD and NMD is related to translation... But should you consider Y14, MAGOH etc. as well in this review (they do have a role in the nucleus by definition)?

It is another example of the CTA paralog with nuclear localization.

522: again, as in the case of “eEF1A2 isoform” before: the phrase “the eIF4G1 paralog” most likely means eIF4G2 and not eIF4G1.

Corrected

537: “polyA sites” – it is probably “polyadenylation sites” what the authors mean.

Corrected

548: “eIF4E-4ET” looks strange: it is likely eIF4E transporter, which is usually called 4E-T or eIF4ENIF.

Corrected

552: “mediateD”.

Corrected

559: “aninhibitor”.

Corrected

565: “mG cap”

Corrected

580-587, 601-605: I suggest moving these paragraphs to the appropriate section (#2).

Corrected

650: please add “some” before :other signaling”.

Corrected

668: “the primary nuclear protein” – what does it mean?

Corrected

714: “Our analysis of predicted NLS [407] patterns” – please rephrase this sentence, as now it can be considered as the results of your own analysis described in Ref. 407 (which is not the case).

Corrected

Table 1: please consider my above comment about nomenclature of proteins.

Corrected

Ref.26: please remove <em>/<em>.

Corrected

Round 2

Reviewer 3 Report

The manuscript has been substantially improved and, to my opinion, now it can be accepted for publication. So, no additional review cycles are needed.

However, I would suggest the authors to take a look on the following minor points during the preparation of the manuscript for the very final submission to a journal production office:

359 (it is probably better to use “another”/”one more” instead of “an”, or to add “also” in front of “an”);

503-506 (I still do not understand why RNAP III is mentioned here).

It is also a pity that the authors totally ignored ribosome recycling factors (ABCE1/Rli1p, MCTS1/Tma20p, DENR/Tma22p, and eIF2D/Tma64p), directly involved in translation, while discussed eIF1AD, eIF4AIII and other proteins much more distantly related to protein synthesis. Ribosome recycling is one of the 4 main stages of the translation cycle, along with initiation, elongation and termination; it also plays an important role in translation reinitiation.

However, all these are not very important, so the manuscript can be accepted in its present form.

SD